

**River water quality changes in New Zealand over 26 years (1989 – 2014): Response to land**
**use and land disturbance**
Jason P. Julian*[1,5], Kirsten M. de Beurs[2,5], Braden Owsley[2,5], Robert J. Davies-Colley[3], Anne-
Gaelle E. Ausseil[4]
[1]Department of Geography, Texas State University, San Marcos, TX, USA
[2]Department of Geography and Environmental Sustainability, The University of Oklahoma,
Norman, OK, USA
[3]National Institute of Water and Atmospheric Research Ltd (NIWA), Hamilton, New Zealand
[4]Landcare Research, Palmerston North, New Zealand
[5]Landscape & Land Use Change Institute (LLUCI), http://tethys.dges.ou.edu/main/, USA
*Corresponding author: Jason.Julian@txstate.edu
Abstract
River water quality reflects land use in the catchment (mobilizing diffuse pollution) as well as
point source discharges. In New Zealand (NZ) diffuse pollution vastly outweighs point sources
which have largely cleaned up over many decades. Because NZ has good geospatial data on
physiographic variables, land cover and agricultural statistics, and time series on water quality at
the national scale over several decades, the country is a natural laboratory for investigating water
quality response to land use/disturbance and associated diffuse pollution 'pressures'. We
interpreted water quality state and trends for the 26 years from 1989 and 2014 in the National
Rivers Water Quality Network (NRWQN), consisting of 77 sites on 35 mostly large river
systems with an aggregate catchment amounting to half of NZ's land area. To characterize water



quality pressures, we used multiple land use datasets spanning 1990 - 2012, plus recently-
developed 8-day land-disturbance datasets using MODIS imagery. Current state and directions of
change in visual clarity and nitrate-nitrite-nitrogen provide a particularly valuable summary of
impact, respectively from mobilization of fine particulate matter and soluble nutrients. We show
that the greatest impact on river water quality in NZ over the 1989-2014 period is high-
producing pastures with their high nutrient inputs to support high densities of livestock. While
land disturbance was not itself a strong predictor of water quality, it did help explain outliers of
land use-water quality relationships, especially those with large areas of plantation forest.
Plantation forestry was strongly associated with water quality impacts, particularly on visual
clarity and particulate nutrients when land disturbed for harvesting generated sediment runoff
and nutrient mobilization. In all, our study demonstrates how interdisciplinary combinations of
expertise including geospatial analysis, land management, remote-sensing, and water quality can
advance understanding of broad-scale and long-term impacts of land use change on river water
quality.
1. Introduction

River water quality reflects all that has happened within its catchment, including

geomorphic processes, vegetation characteristics, climate, and anthropogenic land uses (Brierley,
2010). Relationships between water quality and these catchment characteristics are not
straightforward because all of these factors interact over both space and time. For example, if
intensive livestock grazing occurs on steep slopes, surface runoff and consequently river
turbidity is expected to be greater than if grazing occurs on flatter areas. Or if fertilizers are
heavily applied to sandy soils with high drainage density, rivers will likely become eutrophied
over a period of decades due to legacy nutrients slowly leaking to the rivers through



groundwater. The influence of land use on water quality has also been shown to vary among
different climates (Larned et al., 2004). With all of the various types of intensive land uses that
have occurred across diverse landscapes over hundreds of years, rivers with degraded water
quality are now widespread.

Historically, water quality in rivers was managed to meet minimal standards (Baron et al.,

2002). However, in the last decade, a greater emphasis has been placed on maximizing the
ecosystem services provided by healthy rivers, which is driving efforts to improve water quality
(Brauman et al., 2007; Davies-Colley, 2013). Early efforts in developed countries to improve
water quality focused on point-source pollution, particularly wastewater discharges from
factories and treatment plants (Campbell et al., 2004). While the broad-scale reduction in point-
source pollution elevated many water quality variables above minimal standards, most rivers
globally still have water quality impairments due to diffuse pollution – fine sediments, nutrients,
pathogens, toxicants, salts, and other contaminants that are delivered from unknown or many
indistinguishable sources across the catchment (Vorosmarty et al., 2010). Agricultural land uses
are by far the greatest contributors of diffuse pollution, globally (Foley et al., 2005; Vitousek et
al., 1997b); however, the 'intangible' sources of diffuse pollution make it difficult to assign
cause-and-effect relationships (Campbell et al. 2004).

Most studies that have examined relationships between land use and water quality have

used theoretical or numerical models because of the lack of consistent water quality data over
long periods. While this practice can be useful for small catchments where much is known about
its landscape, land-water relationships are complex with interdependencies, feedbacks, and
legacy effects. Empirical studies can shed light on some of these complexities, but they are only
useful for their particular catchments and may have limited generality or transferability.





Comparisons of many diverse catchments is probably most useful to advance understanding of
broad-scale land-water relationships.
One of the most comprehensive empirical riverine studies to date on land use-water
quality relationships has been Varanka and Luoto's (2012) study of 32 boreal rivers in Finland.
They analyzed five water quality variables over 10 years as a function of a suite of
physiographic, climate, and land use variables. A similar study was conducted on many of the
same rivers in Finland, but with a more sophisticated temporal analysis (Ekholm et al., 2015).
And several other studies have used this same river water quality dataset to investigate
environmental drivers. Like Finland, New Zealand (NZ) has an extensive river water quality
monitoring network, which has allowed many studies on river water quality state and trends
(Smith et al., 1996, 1997; Scarsbrook et al., 2003; Scarsbrook, 2006; Ballantine and Davies-
Colley, 2014) and effects of land use (Davies-Colley, 2013; Larned et al., 2004, 2016).
Here, we use NZ as a case study to illustrate long-term relationships among land
management, geomorphic processes, and river water quality. NZ provides a particularly valuable
case study because: (1) it has had one of the highest rates of agricultural land intensities over
recent decades and thus serves as a potential indicator for some developing countries that are
also increasing agricultural intensity; (2) it has one of the longest comprehensive national water
quality datasets in the world; and (3) it is physiographically-diverse. We examined monthly data
for a suite of water quality variables over a 26-year period for 77 very diverse catchments. We
then compared these states and trends of river water quality to landscape data that characterized
the geomorphology, soil properties, and hydro-climatology of these catchments. We also
assessed temporal changes in land cover/use, livestock, and land disturbance over our study
period and compared these to temporal changes in water quality variables. Altogether, these



analyses illustrated coincident spatiotemporal patterns in land use and water quality in NZ rivers
over a quarter of a century. Most of our analyses were performed at the catchment scale because
it integrates the spatiotemporal changes that are reflected in our water quality measurements, it is
the appropriate scale to analyze diffuse pollution, and it is the most appropriate spatial
management unit (Howard-Williams et al., 2010).

2. Study area

New Zealand, (*Aotearoa* "Land of the long white cloud" in the language of indigenous

*Maori* people), is a small island nation (~268,000 km$^2$) located between the South Pacific Ocean
to the east and the Tasman Sea to the west. Its two main islands (North Island and South Island)
are located between 34° and 47° S latitude. Being located on the active boundary between the
Australian and Pacific Plates, NZ's geology and geomorphology are very diverse, including
active volcanoes, karst regions, a range of high fold mountains (the Southern Alps), large coastal
plains, and rolling hills across both hard- and soft-rocks. Being stretched latitudinally, with
nowhere more than about 150 km from the sea, between two major ocean waters combined with
its topographic variability, NZ also has a diverse climate with regional extremes, including sub-
tropical in the far north, temperate in the central North Island, extremely wet on the western side
of the Southern Alps (up to 10 m annually), and semi-arid in the rain shadow to the east of the
Southern Alps.

New Zealand is the last major habitable landmass to be settled by humans. Eastern

Polynesians first arrived around 1300 AD (Wilmshurst et al., 2008). Europeans first arrived in
the late-1700s, but large-scale settlement did not begin until the 1840s. Broad-scale agriculture
spread shortly after and has been intensifying since. While we address land use changes at the





national scale in this study, our water quality analyses focus on 77 diverse catchments across NZ
(Fig. 1), which cumulatively cover about half of NZ's land area.

3. Methods
3.1. Water quality data

Water quality data was obtained from NZ's National Rivers Water Quality Network

(NRWQN), operated and maintained by the National Institute of Water & Atmospheric Research
(NIWA). This network represents one of the world's most comprehensive river water quality
datasets: thirteen water quality and two biomonitoring variables have been measured monthly
(via in situ measurements and grab samples), with supporting flow estimation, from 1989-2014
at 77 sites whose catchments cumulatively drain approximately half of New Zealand's land
surface (Davies-Colley et al., 2011). Further, this dataset has been operationally stable
throughout its history, which allows us to calculate trends over this period. For this study, we
focused on eleven water quality variables and their coincident flow (Table 1). We did not
analyze ammoniacal nitrogen ($NH_4$) because early $NH_4$ samples were biased high by laboratory
contamination (Davies-Colley et al., 2011).

All water quality variables, except water temperature ($T_w$), were flow-normalized (for

each site separately) in JMP® Pro (v 11.2.1) with local polynomial regression (LOESS) using a
quadratic fit, a tri-cube weighting function, a smoothing window (alpha) of 0.67, and a four-pass
robustness to minimize the weights of outliers (Cleveland and Devlin, 1988); where, flow-
adjusted value = raw value – LOESS value + median value. With LOESS, there is no assumption
about the water quality variable's relationship with flow. For example, although visual clarity
usually decreases systematically with increasing flow (Smith et al., 1997), algae blooms at low


flows can sometimes reduce clarity. LOESS also allowed us to examine relative water quality
changes over long periods.

3.2. Physiographic data

Water quality metrics and trends were compared to a suite of landscape variables (Table

2). Catchment morphometrics (area, slope, ruggedness) were obtained from a 30-m digital
elevation model (DEM) that we rescaled from the 25-m DEM produced by Landcare Research.
This DEM was interpolated from 20-m contours of the national TOPOBASE digital topographic
dataset supplied by Land Information New Zealand (LINZ; scale: 1:50,000). Catchment area ($A$)
is the drainage area (in km$^2$) above the NRWQN station, derived using Arc Hydro tools in
ArcGIS 9.3.1 in combination with the River Environment Classification (REC, v2.0) produced
by NIWA. Mean catchment slope ($S_c$) was derived from the same software package, using a 3x3
cell window. We defined ruggedness ($R_r$) as the standard deviation of the 30-m slope grid for
each catchment (sensu Grohmann et al., 2011). Drainage density ($D_d$) was calculated from the
ratio of the total length of REC streams over catchment area (in km/km$^2$).

Soils data was obtained from the 1:50,000 Fundamental Soils Layers (FSL), which is

maintained by Landcare Research. Methods and data descriptions for this soils database are
described in Webb and Wilson (1995) and Newsome et al. (2008). Catchment-scale soil
variables (mean value across catchment) that we included in our analysis for being expected to
be related to water quality were: soil depth ($Z_s$), percent of catchment dominated by silty and
clayey surface soils ($SC\%$), soil pH ($pH_s$), cation exchange capacity ($CEC$), organic matter
percentage ($OM\%$), and phosphate retention ($P_{ret}$). Phosphate retention is a measure (in %) of the




amount of phosphate that is removed from solution by the soil via sorption. Thus, soils with high
$P_{ret}$ have low P-availability for plant growth.
Median annual precipitation ($MAP$), median annual temperature ($MAT$), and median
annual sunshine ($MAS$) averaged across each catchment was obtained from NIWA's National
Climate Database, which contained 5-km gridded daily weather data (Tait and Turner, 2005).
Our values for these three variables represent the median annual precipitation (total mm/y),
temperature (mean °C), and sunshine (hours/y) for the period 1981-2010. Relative water storage
($RWS$) was calculated as the proportion of the annual water yield stored in lakes and reservoirs.
Reservoir/lake storage was obtained from the Freshwater Ecosystems of New Zealand (FENZ)
Database, described in Snelder (2006). The last hydro-climatological variable we included in our
analyses was the median discharge ($Q_{50}$), which was calculated from the NRWQN 'flow
stamping' at times of water quality sampling from 1989-2014.

3.3 Land use and disturbance data
There are two national land use datasets for New Zealand. The Land-Use and Carbon
Analysis System (LUCAS) was developed by the NZ Ministry for the Environment (MfE, 2012)
for reporting and accounting of carbon fluxes and greenhouse gas emissions, as required by the
United Nations Framework on Climate Change and the Kyoto Protocol. Accordingly, LUCAS
uses 1990 as its reference year and maps land use for 2008 and 2012 as well for 12 classes. The
Land Cover Database (LCDB) was developed by Landcare Research (LCR), with contributions
from MfE, Department of Conservation (DOC), Ministry for Primary Industries (MPI), and
Regional Councils (LCR, 2015). LCDB contains 35 land use classes for 1996, 2001, 2008, and
2012. Both datasets use a minimum mapping area of 1 hectare, and use many of the same data





and methods to map land use. There are however, some key differences in their class
designations and classifications that are important to our analyses: (1) LUCAS includes
Manuka/Kanuka as forest, whereas LCDB designates Manuka/Kanuka as shrub; (2) LUCAS
lumps all post-1989 forests into one class, whereas LCDB differentiates between indigenous and
plantation forests; (3) LUCAS uses a conservative approach to mapping high-producing
grasslands, whereas LCDB uses phenological information to provide more accurate estimations
of high-producing grassland. Because of our focus on (water quality-impacting) plantation
forests and high-producing grasslands, we use the LCDB (v4.1) for our spatial and statistical
analyses. We use LUCAS only to quantify long-term changes from 1990 to 2012, before the
LCDB was initiated in 1996. Table 3 describes the land use classes we used in this research,
which classes are included from both datasets, and the national comparison between LUCAS and
LCDB for 2012.

Livestock numbers for dairy cattle, beef cattle, sheep, and deer (at 1 ha resolution) for

each catchment were derived from maps provided by Ausseil et al. (2013), which is
representative for the year 2011. To assess total livestock impact on land disturbance, we
multiplied each livestock type by its AgriBase stock unit (SU) coefficient: sheep = 0.95 SU, deer
= 1.9 SU, beef cattle = 5.3 SU, and dairy cattle = 6.65 SU (Woods et al., 2006). The total SU for
each catchment was then normalized by catchment area, expressed as stock unit density ($SUD$) in
SU/ha.

Changes in $SUD$ from 1990 to 2012 ($SUD_{2012-1990}$) were assessed using district-level data

from StatsNZ (2015) on total numbers of sheep, deer, beef cattle, and dairy cattle. These
livestock numbers were then aggregated for each catchment and multiplied by their respective
SU coefficient. Stock units per hectare were then compared between 1990 and 2012 to assess





change in livestock impacts in each catchment. For Whakatane and Kawerau Districts, 1993 was
used because 1990 data was unavailable.
Land disturbance (i.e. bare soil) was quantified for all high-producing grasslands ($D_{HG}$)
and plantation forests ($D_{PF}$), as well as the whole catchment ($D_C$) for the period 2000 - 2013.
The methods for calculating disturbance are described in de Beurs et al. (2016). Briefly, MODIS
BRDF corrected reflectance data (MCD43A4) at 463 m spatial resolution and eight day temporal
resolution was used to calculate Tasseled Cap brightness, greenness and wetness based on the
coefficients following Lobser and Cohen (2007). These indices consist of linear combinations of
all seven MODIS reflectance bands to represent general image brightness which is comparable to
albedo, image greenness which is comparable to the better known vegetation indices such as
NDVI and EVI, and image wetness which is linked to the amount of water captured in the
vegetation, most comparable to normalized difference water indices. Missing pixels were
ignored. We then calculated the mean and standard deviation of each tasseled cap index for each
combination of land cover class (LCR, 2015) and climatic region for each 8-day time period. We
then used these measures to standardize the calculated tasseled cap indices. To determine how
disturbed each pixel was at any point in time, we then calculated the forest and grassland
disturbances. The forest disturbance index is calculated as the standardized brightness minus the
standardized greenness and wetness. The idea is that disturbed forests appear brighter and less
green and less wet than undisturbed forests. The grassland index is the negative sum of all
indices, indicating that disturbed grasslands appear darker, less green and less wet than
undisturbed grasslands.

3.4 Statistical methods



We used nonparametric Spearman rank correlation coefficients ($r_s$) instead of actual

values to look at relationships between variables, because many of the relationships are

curvilinear. Statistical significance was taken to be an alpha of 0.05. Bivariate comparisons

between all variables (Tables 1-3) were performed to explore for associations and identify

correlated variables before later multivariate analyses. Median values (from the 26-y monthly

time-series) for water quality variables at each site were used when compared to physiographic

and land use variables of their corresponding catchment. Stepwise regression was then used to

rank-order the relative contributions of multiple landscape variables associated with each major

water quality variable. Stepwise regression was used because it accounts for correlations among

the independent landscape variables. The order of variables in the stepwise regression model and

the sign of their coefficient (proportional [+] vs. inverse [-]) provides an objective measure of the

contribution of each landscape variable to river water quality. The level of entry into the model

was set to p = 0.05. All the above statistical analyses were performed in JMP® Pro (v 11.2.1).

Temporal trends in water quality (1989 – 2014) and disturbance (2000 – 2013) data

were assessed with the seasonal Kendall test which was corrected for temporal autocorrelation

using the rkt R package; missing values were ignored. We also calculated the Seasonal Kendall

slope estimators (SKSE) using the same R package. Because some NRWQN sites had multiple

measurements in some months, a few records (no more than five) were removed from each site

in order to ensure 12 monthly values for each year for the SKSE test. There were also occasional

missing values for some variables throughout the time-series, particularly in the early years. Of

particular note, there were no *TN* values for 1994 as a result of contamination by leaking

ammonia refrigerant during storage of frozen subsamples. HV1 did not have data for 18 months

from 2012-2014.





In order to make trend comparisons among sites and derive an estimate of percent change
per year, we normalized SKSE values by dividing them by the raw data median to give the
relative SKSE (RSKSE) in percent change per year (Smith et al., 1996). Given that water
temperature ($T_w$) uses an arbitrary scale in °C, we only report SKSE values for this variable. We
also used the trend categories of Scarsbrook (2006): (1) no significant trend – the null hypothesis
for the Seasonal Kendall test was not rejected ($p > 0.05$); (2) significant increase/decrease – the
null hypothesis for the Seasonal Kendall test was rejected ($p < 0.05$); and (3) 'meaningful'
increase/decrease – the trend was significant, and the magnitude of the trend (RSKSE) was
greater than 1% per year. According to Ballantine and Davies-Colley (2014), a 1% change per
year translates to slightly more than 10% change per decade (due to compounding), a rate of
change that is easily detectable and observable.

4. Results
4.1. Physiographic characteristics
The 77 NRWQN catchments were physiographically diverse in terms of morphometric,
soil, and hydro-climatological variables (Table 4; Supplement Table 1). Most notable with
regards to its direct influence on runoff and water quality was median annual precipitation
(*MAP*), which ranged from 533 to 7,044 mm/y. When combined with the wide range of
catchment areas (*A*), median discharge ($Q_{50}$) varied over three orders of magnitude, from 0.4 to
515 m³/s, and annual water yield from 103 to 3,475 mm/y. In terms of soil, about a quarter of the
catchments had very sandy surface soils (*SC%* < 10) and a quarter had fine-textured soils (*SC%*
> 70). Phosphate retention ($P_{ret}$), an important variable for fertilizer management and





consequently water quality, was particularly high (>57%; 10[th] percentile) for catchments HM2,
HM5, HM6, WA1, WA2, WA3, and WN5.

Several physiographic variables (Table 2) displayed strong latitudinal trends from North

to South ($r_s$): $MAT$ (-0.83), $MAS$ (-0.61), $R_r$ (0.58), $Z_s$ (-0.57), and $P_{ret}$ (-0.52). Many of the
physiographic variables were strongly correlated (p < 0.001; Supplement Fig. 1). Notable ones
include ($r_s$): $A$ v $Q_{50}$ (0.89), $S_c$ v $D_d$ (-0.79), $R_r$ v $S_c$ (0.67), $Q_{50}$ v $R_r$ (0.57), $RWS$ v $Q_{50}$ (0.55),
$RWS$ v $A$ (0.54), $R_r$ v $D_d$ (-0.52), $OM\%$ v $Z_s$ (0.47), $MAP$ v $S_c$ (0.47), $Z_s$ v $S_c$ (-0.42), $Z_s$ v $SC\%$
(-0.41), $P_{ret}$ v $pH_s$ (0.40), $MAP$ v $P_{ret}$ (0.39), and $MAT$ v $OM\%$ (0.38). In consideration of these
relationships and perceived importance for water quality (*sensu* Varanka and Luoto, 2012), we
used the following subset of minimally correlated physiographic variables for subsequent
multivariate analyses: catchment slope ($S_c$), silt-clay percentage ($SC\%$), phosphate retention
($P_{ret}$), and median flow ($Q_{50}$).

4.2. Land use change and disturbance

Land use in NZ, like physiography, varied widely; and our 77 catchments captured this

diversity (Fig. 1; Supplement Table 2). Thirteen catchments were dominated (>50%) by non-
plantation forests ($NF$), with one (WN2) containing more than 94%. Thirteen other catchments
were dominated by shrub/grassland ($SG$) that was not intensively managed. The most dominant
land use was grasslands that were intensively managed (hereafter high-producing grasslands;
$HG$), covering the majority of the area for 31 catchments. Together, these three land uses made
up 84% of the catchments' areas. Plantation forest ($PF$) was the majority land use for three
catchments: RO3, RO5, and RO2, all in the volcanic plateau of central North Island. Open water
($OW$) was the majority land use for one catchment (RO1) and relatively high (>10%) for two





others (RO6, DN10). Barren/other (*BO*), which was largely bare rock, was relatively high
(>10%) for 13 mountainous catchments. Urban (*UR*) coverage rarely exceeded 1%, with only
one catchment greater than 2% (WN1). Annual cropland (*AC*) exceeded 1% in 11 catchments,
but never exceeded 8%. Vegetated wetland (*VW*) and perennial cropland (*PC*) were minimal in
all catchments, each rarely exceeding 1%.
In general, non-plantation forest (*NF*), shrub/grassland (*SG*) and barren (*BO*) areas
dominated mountainous catchments with high $S_c$ and low $Z_s$; while high-producing grasslands
(*HG*) dominated most lowland catchments with low $S_c$, high $Z_s$, and high $pH_s$. Like *HG*,
plantation forest (*PF*) mostly occurred on flat areas ($r_s = -0.48$ with $S_c$) with thick soils (0.35
with $Z_s$) that were less acidic (0.31 with $pH_s$). *PF* was also significantly proportional to $P_{ret}$ ($r_s =$
0.24). Given the relative dominance of catchment land use, relationships with physiographic
variables, and potential effects on water quality in NZ rivers (Davies-Colley, 2013; Howard-
Williams et al., 2010), the land use variables used for subsequent multivariate analyses were *NF*,
*SG*, *HG*, *PF*, and *OW*.
Land use change in the 77 catchments from 1990 to 2012 was usually minor (Supplement
Table 2). The greatest change was a 13.4% increase in *PF* in GS1, which was almost entirely
accounted for by a 13% decrease in *SG*. Thirteen other catchments experienced small increases
(3.0 - 6.6%) in *PF*, accounted for by decreases in *SG* or *HG* or both. HM3 and HM4 had the
greatest increases in *HG* at 3.4% and 2.0%, respectively. High-producing grasslands (*HG*) for the
other 75 catchments remained virtually unchanged (< 0.4%) or decreased. WH3 had the greatest
decrease in *HG* at -4.8%. Land use change in other catchments was negligible. Changes in total
stock unit density between 1990 and 2012 ($SUD_{2012-1990}$) were also minor with only two
catchments (AK1 and AK2: both -5.1 SU/ha owing to urban fringe expansion) changing more





than 1.6 SU/ha over this period (Supplement Table 3). Temporal changes in $SUD_{2012-1990}$ for 56
of the 77 catchments were within the range of -1.0 to 1.0 SU/ha.

Although land use and total livestock densities changed little in 1990-2012, livestock

*types* changed considerably for many catchments (Supplement Table 3). The general pattern was
dairy cattle replacing sheep. The number of dairy cattle from 1990 to 2012 increased in 72
catchments, with a mean increase of 0.6 SU/ha for all catchments; while the number of sheep
decreased in all 77 catchments (mean = -0.9 SU/ha). Deer and beef cattle numbers changed little:
0.0 and -0.2 SU/ha, respectively.

When 2011 livestock densities were compared with physiographic variables, the

strongest relationships were found with combined *SUD* of dairy and beef cattle (hereafter
$SUD_{cattle}$; Supplement Fig. 2). $SUD_{cattle}$ decreased strongly with increasing slope, $S_c$ ($r_s$ = -0.79),
but increased with $Z_s$ (0.43), $pH_s$ (0.32), and $P_{ret}$ (0.27). $SUD_{cattle}$ also increased with *MAT*
(0.68) and *MAS* (0.42), but decreased with *MAP* (-0.34). Thus, highest cattle densities were
found in catchments such as WA3 (with the highest $SUD_{cattle}$ at 15.7 SU/ha) that were relatively
flat, warm, sunny, and dry, with deep soils that had relatively high pH and high P-retention.
High-producing grasslands (*HG*) had similar, but less strong, correlations with these same
physiographic variables.

Catchment disturbance ($D_C$) varied widely over both space and time between 2000 and

2013 (Supplement Table 4). The maximum amount of $D_C$ at one time was 35.7% for WN3 on
07-Apr-2003, almost entirely due to bare pastures. $D_C$ exceeded 15% on six other occasions (264
days in total) in this catchment. In general, the North Island (Fig. 2) had a greater extent and
intensity of disturbance than the South Island (Fig. 3). The most intense disturbances occurred as
a result of plantation forest harvests, and these disturbances were on average visible for about 1.5





y up to about 4 y, with exceptions lasting more than 6 y. Indeed, $D_C$ was strongly correlated to
$PF$ coverage ($r_s = 0.51$). The catchment with the highest median $D_C$ (10.5%) was RO3, which
had 69.8% of its catchment in $PF$ and 17.7% in $HG$. Fourteen other catchments had $D_C$ above
5%, and two-thirds of these were dominated by either $PF$ or $HG$.

We also analyzed disturbance of plantation forests ($D_{PF}$) and high-producing grasslands

($D_{HG}$) separately for each catchment. For catchments with at least 21.4-km$^2$ (100 MODIS pixels,
for the sake of statistical robustness) of plantation forest, the mean ($\pm$SD) $D_{PF}$ (from 2000 to
2013) was 10.6 $\pm$ 5.6%. The catchments with the highest $D_{PF}$ were those with low mean annual
precipitation, MAP ($r_s = -0.42$). There were no significant relationships between $D_{PF}$ and any of
the other physiographic variables. For catchments with at least 21.4-km$^2$ of high-producing
grasslands, the mean ($\pm$SD) $D_{HG}$ was 6.0 $\pm$ 6.4%. The catchments with the highest $D_{HG}$ were
those with low mean annual sunshine ($MAS$; $r_s = -0.25$), low mean annual temperature ($MAT$; -
0.30), high catchment slope ($S_c$; 0.25), and high ruggedness ($R_r$; 0.31). The six catchments with
the highest $D_{HG}$ (>15%) all had low phosphate retention ($P_{ret}$; <32%). While it is assumed that
greater densities of livestock lead to greater pasture disturbance, we did not find a proportional
relationship between stock unit density ($SUD$) and $D_{HG}$ across space (i.e. among catchments). In
fact, the highest median $D_{HG}$ was found for catchments with *low SUD* ($r_s = -0.45$). Over time
however, we observed a fairly strong trend ($r_s = 0.50$) of lower $D_{HG}$ with decreasing SUD (-
$SUD_{2012-1990}$). In all there were seven catchments with significant or meaningful decreases in
$D_{HG}$ from 2000 to 2013 (assessed with Seasonal Kendall slope; SKSE), all of which had a
negative $SUD_{2012-1990}$.

4.3 Water quality characteristics and trends





### 4.3.1 Flow relationships

All water quality variables (per site) had strong relationships with flow ($Q$) except water temperature ($T_w$), which instead followed a seasonal pattern. Conductivity ($COND$) generally decreased with $Q$ for most sites, with exceptions being AX1, DN2, DN10, NN5, RO1, RO6, and TK1. For several sites, $COND$ was high for flood flows. Water pH ($pH_w$) decreased with $Q$ for most sites likely due to relatively acidic rainfall, with exceptions being AX3, AX4, RO5, and RO6. Several sites experienced high $pH_w$ during high flows. The typical pattern for dissolved oxygen ($DO$) for most sites was a wide range at low flows, and high flows converging to near 100% $DO$. The exceptions were sites where $DO$ decreased with flow (DN1, HM4, HM5, WH4) and lake-fed sites where $DO$ was high (>90%) for virtually all flows (AX2, AX4, DN4, D10, RO1, RO6, TK4).

Visual clarity ($CLAR$) had a strong (mean $r^2$ of 0.53 among all sites) exponential-decay trend with flow for almost all sites, as has been reported previously (Smith et al., 1997). Four sites, all lake-fed, had their highest $CLAR$ for intermediate flows (DN10, RO1, RO6, HM3). Of these four, the first three had high $CLAR$ (> 2m) for virtually all flows. Turbidity ($TURB$) had generally the opposite trend of $CLAR$ (as could be expected given the inverse relationship of these variables), and increased near-linearly with $Q$ (albeit with more scatter than $CLAR$). Several of the lake-fed sites had relatively low $TURB$ at *high* flows (AX1, AX2, DN1, DN10, RO1, RO2, RO6). Colored dissolved organic matter ($CDOM$) generally increased with flow as has been reported previously by Smith et al. (1997); the lake-fed sites of RO1, RO2, and RO6 were exceptions. CDOM was sometimes low during floods, likely due to a dilution effect.

Total nitrogen ($TN$) generally increased with $Q$, but with a high degree of scatter, for almost all sites. The exceptions were AX1, AX2, AX4, DN10, RO1, RO6, and TK4, where $TN$





was low for all flows, usually less than 100 mg/m$^3$. The trends of oxidized nitrogen ($NO_x$) with
flow varied widely among the sites. For many sites (26/77), $NO_x$ increased with $Q$, usually with
a positive logarithmic trend (i.e. asymptotes at high flows) due to dilution effects at high flows.
A couple sites displayed a concave upward parabolic trend where $NO_x$ concentrations were
lowest for intermediate flows and high for both low and high flows (CH2, DN6), which we were
unable to explain but is likely due to source of flow. Total phosphorous ($TP$) generally increased
with $Q$ at 73 of the sites, reflecting mobilization of suspended matter (containing P) with $Q$.
Exceptions were the lake-fed sites of DN10, RO1, and RO6, where $TP$ was low for all flows,
usually $\leq$ 10 mg/m$^3$. At the lake-fed site of RO2, $TP$ actually decreased with $Q$. Dissolved
reactive phosphorus ($DRP$) generally increased with $Q$ for most sites; however, there were many
exceptions. Twenty sites had no detectable trend with $Q$ ($r^2 < 0.10$). $DRP$ actually decreased with
$Q$ at four sites (HM5, RO2, TU2, WA3).

4.3.2 Catchment characteristics
Median monthly values of water quality variables for the 77 catchments ranged widely
(Table 5; Supplement Table 5). Some rivers had exceptional water quality all around, while
others had either current issues with multiple variables or worsening temporal trends (assessed
with SKSE from 1989 to 2014; Table 6). Because of the dependence of water quality on flow,
we first assessed temporal trends in $Q$. Only two catchments had significant increases in $Q$
(AX4, WH4), with the latter also being 'meaningful.' Three catchments had significant decreases
in $Q$ (HM3, HM5, TU2) and five others also had 'meaningful' decreases in $Q$ (CH2, GY4, HM4,
RO3, RO4).





Water temperatures ($T_w$) were not particularly high for any of the catchments; however,
21 rivers had significant increases in $T_w$, possibly the signature of climate change. The highest
rates of $T_w$ increase (0.04°C/y < SKSE < 0.08°C/y) were for large alpine rivers in the central
South Island covered mostly by shrub/grasslands (TK3, TK4, TK6, AX3). Because of its strong
latitudinal trend (stronger than any land use effect), $T_w$ was not analyzed further. Dissolved
oxygen ($DO$) was close to 100% for most catchments, but was particularly low (<90%) for two
catchments: RO2 which was affected by discharge from a large pulp mill at Kawerau, and AK2
which is on the Auckland fringe and thus affected by various peri-urban activities. $DO$ was very
high (>110%) for one catchment (HV2) due to supersaturation from high periphyton in this
nutrient-enriched river. Temporal trends in $DO$ from 1989 to 2014 were relatively minor
(RSKSE < 0.5%/y), except RO2 which had a significant increase (RSKSE = 0.7%/y) attributable
to progressive improvements in treatment of organic waste from its large pulp mill. Conductivity
($COND$) was low (<115 µS/cm) for all South Island catchments and varied considerably for the
North Island (54-528 µS/cm). Most catchments (52/77) experienced significant or 'meaningful'
increases in $COND$ from 1989 to 2014. Water pH ($pH_w$) was neutral to alkaline for all rivers,
which have been described as calcium-sodium bicarbonate waters by Close and Davies-Colley
(1990), and only displayed minor changes (RSKSE < ±0.1%/y) over the 26-year study period.
Median visual water clarity ($CLAR$) was exceptionally high (>5 m) for seven catchments
and very low (<1 m) for 22 catchments. Since 1989, $CLAR$ improved in almost half of the rivers,
and worsened in 4 rivers (Table 6; Supplement Table 5). $TURB$ was strongly inversely
proportional to $CLAR$ ($r_s$ = -0.97) and generally followed opposite trends of $CLAR$. However,
fewer of its trends were significant and it had a disproportionally large number of 'meaningful'
increases (17 catchments compared to only 2 'meaningful' decreases in $CLAR$). $CDOM$ was low



for most of the rivers, with only five catchments greater than 2.0 m$^{-1}$. Nineteen of the catchments
experienced significant or 'meaningful' decreases in CDOM since 1989. Only one catchment
had a 'meaningful' increase in *CDOM* (TK3).
Total nitrogen (*TN*) was high (>250 mg/m$^3$) for more than half of the catchments, with
the vast majority (30/39) of these being lowland catchments (<150 m in elevation). Most of these
catchments also had high $NO_x$. Thirty-three catchments had significant or 'meaningful' increases
in *TN* from 1989 to 2014, while only five had significant or 'meaningful' decreases in *TN* (Table
6). $NO_x$ had a similar number of increasing temporal trends, but also had 'meaningful' decreases
for 12 catchments. Total phosphorous (*TP*) followed a similar geographical pattern as *TN*.
Eighteen of the 23 catchments with high *TP* (>30 mg/m$^3$) were lowland catchments. Most of the
catchments with high *TP* (18/23) also had high *DRP* (>9 mg/m$^3$). Seventeen catchments had
'meaningful' increases in *DRP*, compared to only three with 'meaningful' decreases. There was
more of a balance in temporal trends of *TP*, with eight 'meaningful' increases and seven
'meaningful' decreases.
In addition to the expected correlations between *CLAR* and *TURB,* and among the
nitrogen and phosphorous constituents, several other significant relationships existed among the
water quality variables (Supplement Fig. 3). *TP* was correlated with *CLAR* ($r_s$ = -0.77), *TURB*
(0.73), *TN* (0.71), $NO_x$ (0.61), *CDOM* (0.62), and *COND* (0.65). *DRP* was also correlated with
*TN* (0.71), $NO_x$ (0.65), and *CDOM* (0.58). *CDOM* was correlated with *TN* (0.63). Finally, *COND*
and $T_w$ were correlated (0.67). Taking into consideration this broad multicollinearity, we focus
our multivariate analyses on several key water quality variables, particularly those that
experienced the most changes from 1989 to 2014 (Table 6): *CLAR*, *TN*, $NO_x$, *TP*, and *DRP*.



4.4 Water Quality relationships with physiography, land use, and disturbance

There was a predictable relationship between catchment area ($A$) and $Q_{50}$ ($r_s = 0.89$; all

following parentheses in this section are $r_s$ unless specified), and $CLAR$ generally decreased with
$A$ (-0.37). Except for $TURB$ (0.32), no other water quality variables had significant relationships
with catchment area. Several water quality variables correlated with catchment slope ($S_c$),
including: $TN$ (-0.72), $TP$ (-0.63), and $DRP$ (-0.65), meaning N and P concentrations were
relatively high in lowland (low slope) catchments. $DRP$ (0.65) and $TP$ (0.61) were directly
proportional to mean annual temperature ($MAT$), but this association probably arises because the
highest phosphorus values occurred mainly in lowland catchments and some of the northernmost
catchments, temperature being strongly correlated with altitude and latitude. $DRP$ also had a
(counterintuitive) significant relationship with soil phosphate retention, $P_{ret}$ (0.35). No other
strong physiographic relationships emerged from our analyses.

The strongest relationships between water quality and land use (Table 7) included high-

producing grasslands ($HG$), which had strong positive relationships with several water quality
variables except $CLAR$ which decreased as $HG$ increased. The lesser-managed shrub/grasslands
($SG$) had generally opposite relationships with water quality, but note that $SG$ did not have
significant relationships with $TURB$ or $CLAR$. Non-plantation forest ($NF$) followed the same
trends as $SG$, but had fewer significant relationships with water quality. Plantation forest ($PF$),
on the other hand, followed the same trends as $HG$, with poorer water quality being associated
with greater coverage of $PF$; although correlations were not as strong as $HG$. $CDOM$, $DRP$, and
all N-constituents had significant negative correlations with open water ($OW$), meaning that
water quality improved with greater $OW$ coverage, plausibly due to entrapment of fine sediment
and nutrients.





Water quality was correlated with all stock unit density (SUD) metrics (Table 7;
Supplement Fig. 4), except deer ($SUD_{de}$) which only had relatively weak relationships with $TN$
and $NO_x$. The nutrients and CDOM had the strongest correlations with $SUD_{cattle}$, which includes
both dairy and beef cattle. $COND$, $CLAR$, and $TURB$ had the strongest (slightly) correlations
with $SUD_{be}$. Overall, degraded water quality was strongly associated with high livestock
densities, even stronger than coverage of high-producing grasslands.
No significant correlations between water quality and total catchment disturbance ($D_C$)
were found; however, there were significant associations when disturbance was isolated by high-
producing grasslands ($D_{HG}$) and plantation forest ($D_{PF}$; Table 7). Unexpectedly, $CLAR$ and
$TURB$ were not correlated to $D_{HG}$, and surprisingly, the rest of the water quality variables had a
significant *inverse* relationship with $D_{HG}$. Conversely, $CLAR$ was the only water quality variable
correlated to plantation forest disturbance, $D_{PF}$ ($r_s$ = -0.27). Some interesting results emerged
when temporal trends in water quality (via SKSE) were assessed for catchments with high
disturbance. Of the 15 catchments with $D_c$ greater than 5%, six had 'meaningful' increases in
$TURB$ (RO3, HM4, RO6, WA6, HV6, HM2; all in North Island); while only one (HV5) had a
'meaningful' decrease in $TURB$. Most of these 15 catchments also experienced significant
increases in $TN$ (9 catchments; 7/9 also 'meaningful') and $NO_x$ (10 catchments; 8/10 also
'meaningful'). Interestingly, $TP$ and $DRP$ significantly increased in only two of these highly
disturbed catchments.

4.5 Multivariate water quality relationships
In order to build on the above correlation analyses, the water quality variables of $CLAR$,
$TN$, $NO_x$, $TP$, and $DRP$ were each assessed in a multivariate stepwise regression, using the





following ten physiographic and land use independent variables: $S_c$, $SC\%$, $P_{ret}$, $Q_{50}$, $NF$, $SG$,
$HG$, $PF$, $OW$, and $SUD_{cattle}$ (Table 8). The residual plots for all five water quality variables met
the assumptions of normality and linearity, but displayed heteroscedasticity with wide scatter for
high values. $CLAR$ was correlated to $-HG$, followed by $+OW$, $-Q_{50}$, and $-PF$, where signs
represent whether the relationship is positive (+) or inverse (-). Thus, water clarity was
predictably lower for larger rivers that drain larger areas of high-producing grasslands and/or
plantation forests, but improved with increased open water coverage (Fig. 4).

The combined stock unit density for beef and dairy cattle ($SUD_{cattle}$) was the primary

predictor for all four nutrient variables, with $TN$, $TP$, and $DRP$ also being proportional to
plantation forest coverage ($PF$; Table 8). Coverage of high-producing grasslands ($HG$) and silt-
clay surface soils ($SC\%$) were also proportional factors for TN. In sum, land use was the primary
and secondary predictor for all five water quality variables (Fig. 4).

5. Discussion
5.1 River water quality states and trends

We found a wide range of water quality across NZ rivers (Table 5), with drastic

differences between upland and lowland rivers, distinguished by the 150 m elevation threshold.
For example, visual water clarity ($CLAR$), which is often used as a 'master variable' for overall
water quality (Davies-Colley et al., 2003; Julian et al., 2008), was high for upland rivers (mean =
3.2 m), with only two [alpine glacial flour-affected] rivers below the ANZECC (2000) guideline
of 0.6 m (CH3, AX3). Many of the upland rivers (7/33) had very high water clarity (> 5 m),
including one of the clearest non-lake-fed rivers in the world − Motueka River (NN2) with a
median $CLAR$ of 9.8 m. The lowland rivers, in contrast, had a mean $CLAR$ of 1.2 m, with 17





(39%) below the ANZECC guideline of 0.8 m. Note that these ANZECC (2000) guidelines,
which are statistical derivations (i.e. 20th-percentile of the first decade of the NRWQN record for
'reference' sites), are merely 'trigger values' that when exceeded trigger a management response
to protect ecosystem health (Hart et al., 1999). Although these 'trigger values' are not effects-
based standards (which would be difficult to define for the wide variety of NZ ecosystems), they
do provide a useful reference for comparing water quality states and trends. Save for a few
borderline exceptions, the same sites that were below visual clarity guidelines also exceeded the
turbidity trigger values of 4.1 and 5.6 NTU for upland and lowland rivers, respectively.

Nutrient concentrations in NZ rivers also varied widely (Table 5), again with high

concentrations typically in lowland catchments and low concentrations in upland catchments.
Nine of the ten catchments with the highest TN (>740 $mg/m^3$) were lowland catchments. In all,
13 lowland catchments exceeded the ANZECC *TN* guideline of 614 $mg/m^3$ and 8 upland
catchments exceeded the guideline of 295 $mg/m^3$. Almost three quarters of these catchments
(15/21) also exceeded the $NO_x$ guideline of 444 $mg/m^3$ (lowland) and 167 $mg/m^3$ (upland). There
were a similar number of sites exceeding guidelines for *TP* (33/26 $mg/m^3$ for lowland/upland)
and *DRP* (10/9 $mg/m^3$ for lowland/upland), each with at least 20 and most of these were
corresponding. Our results on the state and trends of the 77 NRWQN catchments generally
accord with earlier NRWQN studies (e.g. Ballantine and Davies-Colley, 2014) and a recent
publication by Larned et al. (2016), which analyzed water quality states and trends for 461 NZ
river sites for the period 2004-2013.

Based on ANZECC (2000) trigger values, we have organized the catchments into four

classes (Fig. 5): I. clean river with high visual water clarity (*CLAR*) and low dissolved inorganic
nutrients (DIN); II. sediment-impacted river with low *CLAR* and low DIN; III. nutrient-impacted



river with high *CLAR* and high DIN; and IV. sediment- and nutrient-impacted river with low
*CLAR* and high DIN. Note that the term 'sediment-impacted' is a connotation for total suspended
solids (TSS), which includes organic matter as well. In agriculture-dominated catchments, both
mineral sediment and particulate organic matter can greatly increase TSS (Julian et al., 2008).
We use *CLAR* as a preferred metric for suspended matter because TSS is not routinely measured
in the NRWQN (or other monitoring networks) while *CLAR* correlates strongly to TSS ($r = -$
$0.92$), and better than *TURB* ($r = 0.87$) (Ballantine et al., 2014). Further, *CDOM* in NZ rivers is
low with minimal impact on *CLAR*. We use $NO_x$ as our preferred metric for DIN because it is
least affected by suspended sediment and soil properties (compared to *DRP*). However,
catchments that exceed ANZECC guidelines for *DRP* are indicated in Fig. 5 by grey-filled
markers.

When this classification is combined with the SKSE trend analyses (Table 6), we obtain a

clear picture of the current and potential state of NZ rivers (Fig. 5). Before individual rivers are
discussed (next section), we first point out key differences between the upland and lowland
catchments, which will later be placed within the context of physiography and land use. Most
obvious, and consistent with the findings of Larned et al. (2004), was that lowland rivers were
much more degraded, particularly by sediment. More than a third of the lowland catchments
were either Class II or IV (17/44); whereas, only two upland catchments were Class II. None of
the upland catchments were Class IV, and more than two-thirds were clean rivers (Class I). Both
types had a similar number of nutrient-impacted rivers (Class III). Another major difference is
that all but three of the upland catchments are far from class boundaries, meaning that they are
relatively stable in terms of water quality. Further, almost all of the upland catchments that have
had significant increases in $NO_x$ were already nutrient-impacted. Conversely, many of the





lowland catchments are very close to class boundaries, with most of these having recently
changed classes or likely crossing over in the near future. Particularly concerning is that almost
half of the lowland rivers (19/44) are currently experiencing 'meaningful' increases (>1% per
year) in $NO_x$, *DRP*, or both. The other striking trend is that many of the lowland rivers are
becoming clearer, with 18/44 experiencing 'meaningful' increases (>1% per year) in *CLAR* –
which, plausibly, has been attributed to increasing riparian fencing to exclude cattle from
channels (Davies-Colley, 2013; Ballantine and Davies-Colley, 2014; Larned et al., 2016).
While clearer rivers are seen as an improvement in water quality; when combined with
increasing nutrients, warmer water, and lower flows, the perfect recipe for toxic algae blooms is
created. Only recently has the widespread problem of toxic algae blooms in NZ rivers been
evidenced (Wood et al., 2015; McAllister et al., 2016), and our results indicate that this problem
could worsen given the increasing trends we found in water temperatures, DIN, and most
influential in our opinion, water clarity. Eutrophication and global warming receive the most
attention when it comes to degraded water quality, but rivers have increasingly become light-
limited (Hilton et al., 2006; Julian et al., 2013) such that when clarity improves in warm,
nutrient-rich rivers, algae can proliferate. Particularly problematic for NZ is that its lowland
catchments, which are warmer (mean median $T_w$ of 13.6 v 10.8 °C for upland rivers), have much
greater DIN, and have longer water residence times, are the ones becoming appreciably clearer
(Fig. 5). If droughts become more frequent and intense in NZ, toxic algae blooms are also likely
to become more frequent, more widespread, and more problematic. However, this algae response
is complex and depends on a number of interacting factors such that the apparent potential for
increasing algal nuisance might not necessarily be realized in some rivers.



5.2 The role of physiography in dictating land use across NZ

While physiography did not emerge as a significant independent variable in the multivariate analyses (except *TN* with *SC%*), physiography is important because it largely controls the location and intensity of agricultural land uses. The greatest coverages of high-producing grasslands (*HG*) and the highest densities of cattle ($SUD_{cattle}$), the two primary explanatory variables for all five major water quality variables (Table 8), were both found predominantly in flat areas with deep soils located in warm, sunny, and relatively dry climates. Livestock in NZ depend almost exclusively on pasture grasses and thus their productivity is maximized when pasture productivity is maximized. The very large cattle are not well suited for steep slopes, particularly dairy cattle which can weigh more than 500 kg. Deep soils are important because they absorb and hold more water for plant uptake, and are not as susceptible to waterlogging, especially in wetter climates. Year-round and intense grazing is best supported by warm and sunny climates where pasture grasses are highly productive and recover quickly following intense grazing such as strip/rotational grazing which is common in NZ dairy farms.

Another soil property we found to be positively correlated to $SUD_{cattle}$ was phosphate retention ($P_{ret}$). The highest dairy cow densities were found on Allophanic volcanic soils with high $P_{ret}$, likely because these soils respond favorably to P-fertilizer and thus can be managed more intensively. However, soils with high $P_{ret}$ require more P-fertilizer, and thus generally have higher export of *DRP* to rivers. Our finding of a significant positive correlation between these two variables is consistent with this interpretation. Further, we found that high-producing pastures with high $P_{ret}$ had the lowest disturbance ($D_{HG}$), indicating that these intensively managed pastures recover quickly following grazing. In a more comprehensive study of land disturbance across the North Island of NZ, de Beurs et al. (2016) also found that Allophanic soils




had the least disturbance among all soil orders. Where high livestock densities occur in less than
ideal conditions, land disturbance is likely. Our catchment-scale analyses limit our interpretation
of specific situations, but based on our results, field observations and previous remote sensing
analyses, pasture disturbance in NZ will likely be highest during droughts on steep, south-facing
slopes with thin soils being heavily grazed by sheep. Under these conditions, grasses will be
grazed down to bare soil and recover very slowly.
Plantation forests ($PF$) in NZ also correlated with thick soils with relatively high $P_{ret}$ on
flat areas, particularly the pumice soils of the central North Island. The porous nature of the
pumice soils allows them to efficiently hold and regulate nutrients, water, and air; while being
well-draining and resistant to compaction and flooding. Under these conditions, radiata pine (the
dominant $PF$ species in NZ) grows rapidly (mean harvest cycle of 28 y) and can be harvested
year-round. Since 1990 however, many of the $PF$ additions have occurred on steeper slopes in
response to carbon credit incentives, greater economic demand for wood products (PCE, 2013),
and the need for soil erosion control on steep pasture susceptible to land-sliding (Parkyn et al.,

2006).


5.3 Land use and water quality in New Zealand rivers
5.3.1 Land use diversity and effectiveness
Water quality in NZ rivers has been related to regional differences in climate and source-
of-flow (Larned et al., 2016); however, we focus here on the role of land use because (1) the vast
majority of our catchments were large (only five less than 100 km$^2$) and thus their surface water
quality was likely dominated by catchment characteristics (Julian and Gardner, 2014); (2) the
changes we observed in water quality have been linked to land use globally (Foley et al., 2005;



Vitousek et al., 1997a; Bennett et al., 2001; Walling, 2006); and (3) our results indicate that land
use was the dominant source of diffuse pollutants, and thus influence on spatial and temporal
patterns in river water quality across NZ. Before describing relationships, we would like to first
point out that the 77 NRWQN catchments captured the diversity of land use in NZ, with *NF*, *SG*,
and *HG* (the three dominant land uses of NZ) accounting for 84% of both the 77 catchments and
NZ as a whole. Our empirical study was also an excellent natural experiment in which to assess
the effects of land use on water quality because we had an assortment of dominant land uses
(>50% area) among our catchments (Fig. 1): 24 *HG*, 13 *SG*, 13 *NF*, 2 *PF*, and 25 mixed (i.e. no
single dominant land use).

5.3.2 High-producing pastures and livestock densities

High-producing grassland coverage (*HG*) was the primary explanatory variable for visual

clarity (*CLAR*; Table 8, Fig. 4). *CLAR* in NZ rivers is mostly influenced by mineral and organic
particulates (Davies-Colley et al., 2014). Livestock reduce visual clarity in multiple ways,
especially in NZ where high densities of multiple types of livestock tread year-round on
relatively steep slopes with highly erodible soils vegetated by shallow-root introduced grasses
which are susceptible to destabilization (McDowell et al., 2008). The year-round treading is
particularly important because most NZ regions during winter are very wet with short days,
which increases soil disturbance (pugging and compaction) and slows recovery times. Where
livestock have direct access to rivers, their trampling of riverbanks and instream disturbance is
often the main contributor to reduced *CLAR* (Trimble and Mendel, 1995; McDowell et al., 2008).

The lowland flatter areas in NZ have high *HG* coverage and high cattle stock densities

($SUD_{cattle}$). These lowlands also have high drainage densities – often increased by artificial





drainage. The influence of *HG* on *CLAR* is exacerbated by this interaction of high $SUD_{cattle}$ and
artificial drainage, which explains the high negative correlation between *HG* and *CLAR* (-0.45).
Interestingly, $SUD_{cattle}$ was not an explanatory variable for *CLAR* in the stepwise regression,
which is likely a result of two factors. First, *HG* and $SUD_{cattle}$ are highly correlated, and stepwise
regression does not include secondary variables that are explaining the same proportion of
variance as the primary independent variable. Second, we found that *CLAR* has actually
*improved* in catchments where $SUD_{cattle}$ is high and/or has increased (Fig. 5), which we attribute
to the promotion of riparian fencing across NZ since 2003, when the *Dairying and Clean*
*Streams Accord* was implemented (Bewsell et al., 2007; Howard-Williams et al., 2010). By
excluding (dairy) cattle from channels and riparian zones, the contribution of riverbank and bed
erosion to degraded *CLAR* has been mitigated and reduced over time. Indeed, *CLAR* has been
significantly and meaningfully improving in many of NZ's rivers (Table 6), even those with
increasing $SUD_{cattle}$, albeit from a fairly degraded condition. Of the 34 catchments with
significant increases in *CLAR*, all but 5 had increases in $SUD_{cattle}$ from 1990 to 2012.
Another potential explanation for improved water clarity at numerous sites is the
considerable decrease in sheep density across the NZ landscape. NZ had 57.65 million sheep in
1990. By 2012, that number had been reduced by almost half, to 31.19 million (StatsNZ, 2015).
Although cattle are larger and have a greater treading impact per animal, the much greater
number of sheep means that stock unit density (SUD) may be broadly comparable as regards
environmental impact. Another difference is that sheep are generally placed on steeper, less
stable slopes in NZ, where headwater stream channels are located. Where there are breaks in
slope (even small ones), sheep create tracks of bare soil with their hooves and hillside scars with
their bodies (for scratching and shelter), both of which can enhance soil erosion (Evans, 1997).



Further, cattle (using their tongues) leave approximately half the grass height on the pasture after
grazing; whereas sheep (using their teeth) graze approximately 80% of grass height (down to
bare soil in dire conditions), leaving it exposed to erosion (Woodward, 1998). Considering all
these factors, sheep can have a greater impact on sediment runoff into rivers, and consequently
visual clarity, than suggested by their aversion to water *versus* cattle's attraction to water.
Although not isolated in our analyses, the particulate fractions of *TN* and *TP* have likely been
affected by similar processes as *CLAR* and may follow the same temporal trends (Ballantine and
Davies-Colley, 2014).

While *HG* was also strongly correlated to river nutrient concentrations (Table 7), the

primary explanatory variable for all four major nutrient metrics (Table 8, Fig. 4) was the
livestock density of beef and dairy cattle ($SUD_{cattle}$). The difference between these two
explanatory variables may seem trivial, however the distinction is important if we want to
understand future trends and effectiveness of water quality management strategies. As we
demonstrated, the area of land used for high-producing grasslands (*HG*) has not changed much
since 1990. In fact, it has decreased or stayed virtually the same in all but two of the 77
catchments. Yet, nutrient concentrations have been increasing in many of the rivers (Table 6),
which we attribute to (1) increasing numbers of cattle (mostly dairy) on both *HG* and *SG*, and (2)
legacy nutrients being slowly delivered to the rivers in groundwater. From 1990 to 2012, NZ
approximately doubled its number of dairy cattle, exceeding 6.4 million. (StatsNZ, 2015). This
enormous addition to a country that is only 268,000 km$^2$ in area, has been accompanied by more
than 1.426 million tonnes of P-based fertilizers and 335,000 tonnes of N-based fertilizers
annually (1990-2012 mean; StatsNZ, 2015). Of the nutrients consumed by lactating dairy cows,
approximately 79% of N and 66% of P are returned to the landscape in the form of urine and



feces (Monaghan et al., 2007). This results, potentially, in about 260,000 tonnes of N-based and
940,000 tonnes of P-based diffuse pollution. Some of these nutrients will be transported to rivers
during subsequent storms, but a majority will remain (building up) in the landscape to be slowly
added to rivers over decadal time-scales (Howard-Williams et al., 2010).

5.3.3 Plantation forests

All water quality variables were significantly correlated to plantation forest coverage

($PF$; Table 7), with a negative relationship with $CLAR$ (i.e. $CLAR$ was lower for higher $PF$) but
positive for all other variables (i.e. nutrients increased with $PF$). From the stepwise regression,
$PF$ emerged as an explanatory variable for all major water quality variables except $NO_x$ (Table
8), suggesting that its dominant impact on river water quality was from surface runoff. Plantation
forestry activities can add a considerable amount of sediment and nutrient pollution to rivers,
especially during and immediately following harvesting (Fahey et al., 2003; Croke and Hairsine,
2006; Davis, 2005). This harvesting period of maximum soil disturbance usually lasts about two
years (Fahey et al., 2003), but the land cover may remain sparsely vegetated and susceptible to
erosion for several years (but usually not more than 5 y; de Beurs et al., 2016). The greatest $PF$
impact on sediment runoff, and thus potentially $CLAR$, is usually from road sidecast/runoff,
shallow landslides, and channel scouring/gullying (Fahey et al., 2003; Motha et al., 2003;
Fransen et al., 2001).

Rivers receive a pulse of nutrients during the forest harvest, but fertilizers are also

applied at time of re-planting and sometimes routinely to enhance growth (Davis, 2005). Radiata
pine in the pumice soils of the central North Island, the dominant area of $PF$ in NZ, are
particularly responsive to both N- and P-fertilizers and thus likely receive ample supplements.





Like pasture fertilizers, some of these nutrients may be delivered to rivers during intense
precipitation, but there is also a legacy of nutrients left behind. Fertilizers have been applied to
plantation forests in NZ since the 1950s, with an intense period of application in the 1970s
(Davis, 2005). While fertilization rates (tonnes/ha/y) have decreased since 1980, the amount of
$NO_x$ leaving catchments mostly covered in $PF$ has significantly and 'meaningfully' increased
since 1989: RO3 (69.8% $PF$, 3.0%/y RSKSE), RO5 (53.3% $PF$, 1.7%/y RSKSE), and RO2
(42.5% $PF$, 1.2%/y RSKSE). None of these catchments had more than 17.7% $HG$, none had
major increases in $HG$ ($< 0.3\%$), none had major increases in $SUD_{cattle}$ ($< 0.7$ SU/ha), and none
had a significant increase in $D_{PF}$. What the catchments did have in common were all had
gravelly/sandy pumice soils ($< 4.5$ $SC\%$) and all were intensively managed as reported by Davis
(2005) and as indicated by high $D_C$ ($> 6.8\%$). The extended periods of nonvegetated land due to
weed control also increases the amount of nutrients delivered to rivers over the long term (Davis,

2005).


5.3.4 Other land uses
Open water ($OW$) in the form of lakes can remove sediment, nutrients, and $CDOM$ by a
range of processes (Schallenberg et al., 2013;Wetzel, 2001). Consistent with this concept, our
bivariate comparisons showed that catchments with more $OW$ had lower $CDOM$, $TN$, $NO_x$, and
$DRP$ (Table 7). Our multivariate analyses found $OW$ to be an explanatory variable for $CLAR$
(Table 8, Fig. 4), which we attribute to several of the stations with high $CLAR$ being located
downstream of large lakes (AX1, DN10, RO1, RO6). If these 4 catchments are removed, the
relationship between $OW$ and $CLAR$ is not significant. While lakes can improve downstream
water quality, many lakes in NZ, particularly shallow lakes, are experiencing eutrophication and



other water quality issues (Larned et al., 2016; Abell et al., 2011), which can cause regime shifts
(Schallenburg and Sorrell, 2009) and degrade downstream river water quality.

An important land use for nutrient/sediment fluxes that was missing from our analyses

was vegetated wetlands (*VW*), which was a consequence of exceptionally low *VW* coverage in
NRWQN catchments (0.1% on average and a maximum of 2.2%). With such a miniscule
coverage, these residual wetlands do not provide a detectable water quality improvement
function at the catchment-scale (Mitsch and Gosselink, 2000). Historically, wetlands covered
approximately 10% of mainland NZ (Ausseil et al., 2011). This considerable loss (> 90% of pre-
European extent) of wetlands has deprived NZ rivers of many valuable ecosystem services,
especially the filtration/processing of sediment and nutrients (Clarkson et al., 2013; Verhoeven et
al., 2006). If some of these wetlands could be restored, some of the alarming eutrophication
trends we have documented here (Table 6, Fig. 5) could be mitigated. For example, Mitsch et al.
(2001) found that just adding 10% of wetland coverage can reduce up to 40% of the nitrogen
entering receiving waters.

The other important land use missing from our analyses was urban (*UR*), also because

very little of NZ's land area is urban (Table 3), accounting on average for only 0.35% of our
catchment areas (maximum 5.8%). However, urban water management did have major effects on
three of our catchments by reducing *DRP* point sources. The 'meaningful' decrease of *DRP*
(RSKSE = -4.6%/y) in the Manawatu River below Palmerston North (WA9) was due to
progressive improvements in the city's wastewater treatment, particularly after 2008 when a new
main wastewater treatment plant (incorporating P-removal) became fully operational. *DRP* was
also 'meaningfully' reduced (RSKSE = -5.3%/y) for the Ohinemuri River below Waihi (HM6)
when P-removal was added to the Waihi wastewater treatment plant in 2005. And *DRP* for Hutt





River at Boulcott (WN1) was 'meaningfully' reduced (RSKSE = -3.1%/y) with progressive
improvements to the Hutt Valley wastewater treatment, which were completed in 2002. It is
important to note that these point discharge-affected sites were the only ones with meaningful
reductions in *DRP*.

5.3.5 Land disturbance and water quality

So far, we have discussed how land use affects water quality, with a focus on sediment

and nutrient runoff from high-producing grasslands (*HG*) and plantation forests (*PF*). When land
is disturbed (i.e. bare soil), sediment/nutrient mobilization can be enhanced. The most intense
and longest lasting disturbances occurred during plantation forest harvests. Following harvest,
we found that the land remained disturbed for 1-6 years, with a mean of 1.5 years. The overall
mean and median $D_{PF}$ among all catchments was 10%, which means that plantation forestry
leaves large areas of disturbed land at any one time. When this bare land is exposed to intense
precipitation, large quantities of sediment and nutrients can be mobilized into the rivers. This
happened in the Motueka Catchment (NN1) in 2005 when a 50-y storm fell on some recently-
harvested plantation forests. For one of NN1's sub-catchments, the post-harvest disturbed land
caused a five-fold increase in sediment yield compared to pre-harvest events. Following this
event, sediment yields at NN1 were elevated by a factor of 2-3 over the next 3 years (Basher et
al., 2011). Similar sediment erosion events for plantation forests during the post-harvest
disturbance have been documented for other catchments across NZ (Hicks et al., 2000; Phillips et
al., 2005). Because these disturbances only last a few years, they typically do not show up as
temporal trends (via SKSE); however it is possible that they produce enough readily available
sediment to impact water quality for longer periods.





The coincidence of rainstorms on disturbed pasture could have the same effect on
sediment/nutrient runoff if the pasture is connected to the stream network via steep slopes or
adjacent channels/canals (Dymond et al., 2010). Pastures become disturbed from overgrazing,
strip grazing, pugging/soil compaction, tilling/reseeding, cropping/harvesting, or landsliding on
steep slopes. Given the high intensity of grazing management in NZ, all of these are common.
While $D_{HG}$ was lower than $D_{PF}$ on average, $D_{HG}$ had a higher maximum (Table 4).
Spatiotemporal patterns in disturbance between these two land uses were also different (de Beurs
et al., 2016). $D_{PF}$ covered large areas and lasted years at a time; whereas $D_{HG}$ had two patterns:
(1) one related to dairy cattle strip grazing, which were short-lived due to quick recovery times
of grasses in fertilized soils; and (2) more widespread and longer continuous disturbances
occurring on steeper slopes grazed by sheep and beef cattle, particularly following drought
periods. Because our disturbance analyses had a spatial resolution of 463 m, we likely missed
some paddock-scale disturbances. Future work could use Landsat imagery (30-m resolution) to
assess disturbance (sensu de Beurs et al., 2016).
All six catchments with 'meaningful' increases in $D_{HG}$ had large increases in dairy cattle
density 1990-2012 (mean of 1.0 SU/ha across the catchment). Not surprisingly, all six
catchments suffered impacts to water quality. Five of the six had 'meaningful' increases in $DRP$
and three had meaningful increases in $NO_x$ and $TN$. One had a 'meaningful' increase in $TURB$
and three had significant reductions in $DO$. One of these catchments, in particular, may provide a
glimpse into NZ's future if agricultural intensification continues. The Waingongoro River
catchment (WA3) is covered almost entirely by $HG$ (91.2%), with practically all of this land
being used for intensive strip grazing. The $SUD_{da}$ was 15.0 SU/ha in 1990 and increased to 15.4
SU/ha by 2012. The $D_{HG}$ from 2000-2013 had a strong increasing trend of 9.8%/y RSKSE,





associated with the intensification of dairy operations (Wilcock et al., 2009). The result of all this
intensification was that WA3 had 'meaningful' increases in *TP*, *DRP*, and *TN*. The only reason
$NO_x$ did not display a significant trend is because the catchment was already overloaded with a
median river concentration of 1,852 mg/m$^3$. Noteworthy is that these significant trends of
increasing $SUD_{da}$, $D_{HG}$, and nutrients are occurring not only in lowland catchments on the North
Island (WA3, HV2), but also in upland catchments of the North Island (RO6), as well as both
lowland (TK1) and upland (CH3, TK2) catchments on the South Island.
While disturbance was not itself a strong predictor of water quality, it did help explain
outliers of land use-water quality relationships. For example, streams with high DRP (> 20
mg/m$^3$; 10$^{th}$ percentile) had one of two dominant land uses, either plantation forest, *PF* (RO2,
RO3) or high-producing grassland, *HG* (HM5, WA3, WA9, HM4, HM2). The one exception was
RO4, which had relatively low coverage of *PF* (11.2%) and *HG* (2.9%). In fact, RO4 is
dominated by NF (79.1%). Upon closer examination, we found that the small areas of *PF* and
*HG* in RO4 were disturbed frequently. Further, most of the disturbed forestry occurred on steep
slopes and most of the disturbed pastures (practically all sheep and beef) occurred on hilly terrain
adjacent to stream channels. Our high temporal-resolution analyses of disturbance showed that
even though this catchment is mostly indigenous forest, intense disturbances on small
proportions of developed land can have a considerable impact on water quality. RO4 is also
experiencing significant increases in *TURB* and *TP*, as well as a significant decrease in *Q*.
Another outlier example was RO3, which was the only non-*HG*-dominated catchment
with extremely high $NO_x$ (634 mg/m$^3$). RO3 was dominated by *PF* (69.8%), but it had the
highest median disturbance (10.5%) of all catchments. As discussed previously, disturbance in
plantation forests is correlated with harvest frequency and management intensity. In addition to





the many pulses of $NO_x$ from the forest harvests and post-harvest storms over a vegetation-
cleared soil surface, all of the replantings in the N-deficient pumice soils would have been
accompanied by routine N-fertilizer applications (Davis, 2005). And the catchment's well-
drained sandy/gravelly soils meant that this dissolved N was transported to streams without
much attenuation. This catchment also exceeds ANZECC guidelines for $DRP$ and has
experienced meaningful increases in $TURB$, $TN$, and $NO_x$.

We believe that land disturbance and consequently river eutrophication and reduced

visual clarity will continue to worsen in some NZ catchments based on the following. More
plantation forests were planted 1993-1997 (3,810 km$^2$) than any other 5-y period in NZ history
(NZFFA, 2014). With a 28-y mean age of harvest, NZ will experience its greatest coverage and
intensity of forest disturbance around 2025, less than 10 years from now. When combined with
drought and intense storms, the potential for nutrient and sediment mobilization from these lands
into NZ's rivers is high, especially given that approximately 45% of these plantings occurred on
high-producing grasslands (NZFFA, 2014) where many of the legacy nutrients will be exported
to rivers during forest harvest (Davis, 2014). Many of these plantings also occurred on steep
slopes, which exacerbates sediment runoff. If carbon prices continue to stay low, there will be a
high likelihood that many of the harvested forests will be converted to pasture, adding even more
nutrients to NZ rivers (PCE, 2013). Given that the Central Government created a national policy
goal of nearly doubling the export to GDP ratio by the year 2025 (MBIE, 2015), NZ is likely to
see continued increases in livestock density, fertilizer usage, and supplemental feed to support
these extra livestock, all of which will add even more pressure and risks of eutrophication on
NZ's rivers.



**Conclusions**


This study had the overall goal of describing how changes in land use and land
disturbance impact river water quality across broad scales and over long periods. To address this
goal we used a combination of 'brute force' statistical analyses (in terms of hundreds of analyses
using a suite of physiographic, land use, and water quality data for 77 catchments over 26 years)
and careful examination (using multi-resolution data to find patterns and relationships among
these variables). This goal was ambitious and we likely missed some relationships and details of
water quality changes. However, we found empirical evidence for several key relationships
among land use, land disturbance, and water quality, which we now place into a broader
perspective.
The greatest negative impact on river water quality in New Zealand (NZ) in recent
decades has been high-producing pastures that require large amounts of fertilizer to support high
densities of livestock. While this claim has been previously published (Davies-Colley, 2013;
Howard-Williams et al., 2010; and references within), our results and supporting information
show that the relationship between high-producing pastures and water quality is complicated,
being dependent on physiography (particularly soil type), livestock type/density, and disturbance
regime. Dairy cattle receive much of the blame for degraded water quality because of their high
nutrient requirements (Howard-Williams et al., 2010), but beef cattle can also strongly degrade
water quality due to comparable required inputs and grazing on steeper land with a higher
potential for runoff (McDowell et al., 2008). Further, pasture designations/boundaries are
becoming increasingly blurred by modern cattle management, with greater movements of dairy
and beef cattle among pastures, greater use of high-producing pastures for beef, over-wintering
of dairy cattle on beef pastures, and cross-breeding (Morris, 2013). While riparian fencing has no





doubt improved the clarity of NZ rivers, the removal of millions of sheep from steep slopes has
also likely played a role that should be investigated further.

New Zealand is the global leading exporter of whole milk powder, butter, and sheep

products; and NZ's prominence in these industries is likely to continue over the next decade
(OECD/FAO, 2015). Because NZ's economy is heavily dependent on agricultural production,
the agricultural intensification that we have documented since 1990 may be expected to continue,
with greater livestock densities being supported by supplemental feed and fertilizers. Even if best
management practices are adopted to reduce nutrient export to rivers, there is already a half-
century legacy of nutrients distributed across the NZ landscape that will continue to leak to the
rivers. Having an extensive national network like the NRWQN to document and study these
water quality changes is important, but unfortunately the NRWQN is being down-sized at the
time of writing. Less than half of the 77 sites are to be retained by NIWA in a 'benchmark'
network, with 'excess' sites being transferred to regional operation or closed. Although regional
management agencies in NZ conduct much water quality monitoring (e.g. Larned et al., 2016),
the quality (of some) and consistency of their datasets falls short of the NRWQN – which was
also longer-running than all but a very few regional sites.

In response to public concerns on water quality, New Zealand released its National Policy

Statement on Freshwater Management in 2011. Data and evidence-based science is now needed
to support and facilitate limit settings for water quality standards, especially for diffuse pollution
(Duncan, 2014). In their most recent environmental review by the Organisation for Economic
Co-operation and Development (2015), NZ had the highest percent increase (1990-2005) in
agricultural production out of 29 OECD countries, the highest percent increase in N-fertilizer
use, and the $2^{nd}$ highest increase in P-fertilizer use. This massive application of nutrients to the
NZ landscape over our study period is reflected in overall nutrient enrichment of NZ rivers (Fig.
5; Table 6). However due to legacy/lag effects, notably the slow delivery of nutrients to rivers
from land and groundwaters (Larned et al., 2016), the full impact on river water quality will not
be fully appreciated for another several decades (Howard-Williams et al., 2010; Vant and Smith,

2004).


Author contribution
J. Julian designed the study and performed most of the analyses. K. de Beurs developed the
disturbance dataset and performed all trend analyses, both with assistance from B. Owsley. R.
Davies-Colley provided water quality dataset and guidance on its use. A.-G. Ausseil developed
the stock unit density dataset and provided guidance on land use analyses. J. Julian prepared the
manuscript with contributions from all co-authors.

Acknowledgments

The impetus for this research was J.P. Julian's Fulbright Senior Scholar Fellowship,

which was hosted by the National Institute of Water & Atmospheric Research (NIWA) in
Hamilton, NZ in 2012. This work was funded by NASA LCLUC grant NNX14AB77G and NSF
Geography grants #1359970 and #1359948 (Co-PIs Julian and de Beurs). Andrew Tait (Climate
Principal Scientist at NIWA) provided climate data. Agricultural production and other data
essential to this manuscript was collected by William Wright (Landcare Research; LCR). Many
other people in New Zealand provided expert advice, including Suzie Greenhalgh (LCR), Sandy
Elliot (NIWA), Andrew Hughes (NIWA), Deborah Ballantine (then of NIWA), Graham
McBride (NIWA), Murray Hicks (NIWA), David Hamilton (University of Waikato), Les Basher



(LCR), Ian Fuller (Massey University), Roger Young (Cawthron Institute), Rien Visser
(University of Canterbury), and David Lee-Jones (USDA FAS). Support for this project was also
provided by numerous Regional Councils/Districts, including Auckland, Canterbury, Horizons,
Tasman, and Waikato.

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





**Tables**
Table 1. Water quality variables measured by the National River Water Quality Network
(NRWQN) obtained from monthly samples from 1989 to 2014 for 77 catchments.

| Variable | Definition (units) |
| --- | --- |
| $Q$ | Water discharge ($m^3$/s) |
| $T_w$ | Water temperature (°C) |
| $DO$ | Dissolved oxygen (%) |
| $COND$ | Water conductivity ($\mu$S/cm) |
| $pH_w$ | Water pH ($-\log_{10}[H^+]$) |
| $CLAR$ | Horizontal visual water clarity from black disc sighting range (m) |
| $TURB$ | Water turbidity (NTU) |
| $CDOM$ | Colored dissolved organic matter, measured as spectrophotometric absorbance of a membrane filtrate at 440 nm ($m^{-1}$) |
| $TN$ | Total nitrogen ($mg/m^3$) |
| $NO_x$ | Oxidized nitrogen in nitrate and nitrite forms ($mg/m^3$) |
| $TP$ | Total phosphorus ($mg/m^3$) |
| $DRP$ | Dissolved reactive phosphorus ($mg/m^3$) |






Table 2. Landscape variables characterizing the 77 catchments of the National River Water
Quality Network (NRWQN). More details on sources for these data can be found in Methods
section.

| Variable | Definition (units) | Source (resolution/scale) |
|---|---|---|
| **Morphometric variables** | | |
| Area ($A$) | Total catchment area above monitoring site ($km^2$) | National Elevation Dataset (30 m) |
| Drainage density ($D_d$) | Total length of streams per catchment area ($km/km^2$) | River Environment Classification, v2 (1:24,000) |
| Catchment slope ($S_c$) | Mean slope across entire catchment (degrees) | National Elevation Dataset (30 m) |
| Ruggedness ($R_r$) | Standard deviation of catchment slope (degrees) | National Elevation Dataset (30 m) |
| **Soil variables** | | |
| Silt-clay percentage ($SC\%$) | Percentage of catchment surface soils dominated by clayey or silty soils (%) | Fundamental Soil Layers (1:63,360) |
| Soil depth ($Z_s$) | Mean maximum potential rooting depth across catchment (m) | Fundamental Soil Layers (1:63,360) |
| Soil pH ($pH_S$) | Mean pH at 0.2-0.6 m depth across catchment ($-\log_{10}[H^+]$) | Fundamental Soil Layers (1:63,360) |
| Cation exchange capacity ($CEC$) | Weighted mean CEC at 0-0.6 m depth across catchment (cmoles [+]/kg) | Fundamental Soil Layers (1:63,360) |
| Organic matter percentage ($OM\%$) | Weighted mean of total carbon at 0-0.2 m depth across catchment (%) | Fundamental Soil Layers (1:63,360) |
| Phosphate retention ($P_{ret}$) | Weighted mean of phosphate retention at 0-0.2 m depth across catchment (%) | Fundamental Soil Layers (1:63,360) |
| **Hydro-climatological variables** | | |
| Median annual precipitation ($MAP$) | Median annual precipitation averaged across catchment (mm/y) | NIWA National Climate Database (5 km) |
| Median annual temperature ($MAT$) | Median annual temperature averaged across catchment (°C) | NIWA National Climate Database (5 km) |
| Median annual sunshine ($MAS$) | Median annual sunshine hours averaged across catchment (hours/y) | NIWA National Climate Database (5 km) |
| Median discharge ($Q_{50}$) | Median discharge from NRWQN samples during 1989-2014 ($m^3/s$) | NRWQN (catchment) |





| Relative water storage ($RWS$) | Proportion of annual $Q_{50}$ stored in reservoirs/lakes ($m^3/m^3$) | Freshwater Environments New Zealand (1:50,000) |
|---|---|---|
| **Land Use and Land Disturbance variables** | | |
| Land use | Percent of catchment that is occupied by each land use (%); see Table 3 for land uses | Land Cover Database (LCDB, v 4.1), 2001 (1 ha) |
| High-producing pasture disturbance ($D_{HG}$) | Percent of high-producing grasslands within catchment that is disturbed (%), based on aggregate of 463-m pixels within catchment | de Beurs et al., 2016 (463 m; 8-day) |
| Plantation forestry disturbance ($D_{PF}$) | Percent of plantation forestry within catchment that is disturbed (%), based on aggregate of 463-m pixels within catchment | de Beurs et al., 2016 (463 m; 8-day) |
| Catchment disturbance ($D_C$) | Percent of catchment that is disturbed (%), based on aggregate of 463-m pixels within catchment | de Beurs et al., 2016 (463 m; 8-day) |
| Stock unit density ($SUD$) | Catchment-averaged stock unit density for dairy ($_{da}$), beef ($_{be}$), deer ($_{de}$), and sheep ($_{sh}$) in 2011 (SU/ha); subscripts are used to isolate SUD by livestock type | Ausseil et al., 2013 (1 ha) |
| Change in stock unit density ($SUD_{2012-1990}$) | Difference between SUD in 2012 and 1990 (SU/ha) | Statistics NZ (territorial authority) |






Table 3. Land use classification used in this study, aggregated from the LUCAS (v11) and
LCDB (v4.1) land use/cover datasets.

| Class (abbreviation) | Description | LUCAS classes | LCDB classes | 2012 national coverage (%) LUCAS / LCDB |
|---|---|---|---|---|
| Non-plantation forest (NF) | All non-plantation forests ≥ 5m; does not include Manuka/Kanuka | 71 | 68, 69 | 29.2 / 23.9 |
| Plantation forest (PF) | All forests that are planted for the purpose of harvesting | 72,73 | 64, 71 | 7.9 / 7.6 |
| Shrub/Grassland (SG) | All shrubs < 5m and grasses that are not intensively managed | 74, 76 | 41-44, 50-58 | 33.0 / 25.4 |
| High-producing grassland (HG) | High-quality pasture grasses that are intensively managed | 75 | 40 | 21.6 / 33.0 |
| Perennial cropland (PC) | Orchards and vineyards | 77 | 33 | 0.4 / 0.4 |
| Annual cropland (AC) | All annual crops and cultivated bare ground | 78 | 30 | 1.4 / 1.4 |
| Open water (OW) | Rivers, lakes/reservoirs, ponds, and estuaries | 79 | 20-22 | 1.9 / 2.0 |
| Vegetated wetland (VW) | Herbaceous or woody vegetation periodically flooded; includes mangroves | 80 | 45-47, 70 | 0.5 / 0.7 |
| Urban (UR) | Built-up areas, infrastructure, transportation networks, and urban parks/open spaces | 81 | 1-5 | 0.8 / 0.9 |
| Barren/Other (BO) | Bare rock, sand, gravel and other areas not dominated by vegetation; includes mining and permanent ice/snow | 82 | 6-16 | 3.3 / 4.8 |




Table 4. Statistical description of landscape variables for the 77 NRWQN catchments. Refer to Tables 2 and 3 for variable descriptions.

| Variable | Units | Minimum | Median | Maximum | Mean ± SD |
|---|---|---|---|---|---|
| **Morphometric Variables** | | | | | |
| Area ($A$) | km$^2$ | 26 | 1126 | 20539 | 2639 ± 3714 |
| Drainage density ($D_d$) | km/km$^2$ | 1.30 | 1.59 | 2.61 | 1.60 ± 0.16 |
| Catchment slope ($S_c$) | degrees | 3.4 | 15.9 | 30.3 | 16.3 ± 6.8 |
| Ruggedness ($R_r$) | degrees | 3.4 | 10.8 | 15.8 | 10.6 ± 2.4 |
| **Soil Variables** | | | | | |
| Silt-clay percentage ($SC\%$) | % | 0 | 47.3 | 98.7 | 44.0 ± 31.6 |
| Soil depth ($Z_s$) | m | 0.55 | 0.96 | 1.50 | 1.02 ± 0.22 |
| Soil pH ($pH$) | -$\log_{10}$[H$^+$] | 4.8 | 5.6 | 6.5 | 5.6 ± 0.3 |
| Cation exchange capacity ($CEC$) | cmoles [+]/kg | 11.6 | 18.7 | 33.5 | 18.8 ± 4.6 |
| Organic matter percentage ($OM\%$) | % | 2.8 | 6.7 | 23.2 | 7.2 ± 2.9 |
| Phosphate retention ($P_{ret}$) | % | 19.9 | 39.0 | 77.8 | 41.5 ± 12.2 |
| **Hydro-climatological Variables** | | | | | |
| Median annual precipitation ($MAP$) | mm/y | 533 | 1652 | 7044 | 1778 ± 873 |
| Median annual temperature ($MAT$) | °C | 5.0 | 9.9 | 15.1 | 9.9 ± 2.4 |
| Median annual sunshine ($MAS$) | hours/y | 1325 | 1856 | 2116 | 1841 ± 146 |
| Median discharge ($Q_{50}$) | m$^3$/s | 0.4 | 26.0 | 515.0 | 69.6 ± 112.6 |
| Relative water storage ($RWS$) | m$^3$/m$^3$ | 0 | 0 | 29.2 | 1.1 ± 3.7 |
| **Land Use Variables** | | | | | |
| Non-plantation forest ($NF$) | % | 0.1 | 20.5 | 94.1 | 26.7 ± 23.3 |
| Plantation forest ($PF$) | % | 0 | 3.3 | 69.8 | 8.2 ± 12.3 |
| Shrub/Grassland ($SG$) | % | 0.4 | 21.7 | 82.3 | 26.6 ± 20.2 |
| High-producing grassland ($HG$) | % | 0 | 21.6 | 91.2 | 30.9 ± 26.2 |
| Perennial cropland ($PC$) | % | 0 | 0 | 1.3 | 0.1 ± 0.2 |
| Annual cropland ($AC$) | % | 0 | 0.1 | 7.9 | 0.6 ± 1.4 |
| Open water ($OW$) | % | 0 | 0.4 | 25.6 | 1.9 ± 4.3 |



| | | | | | |
|---|---|---|---|---|---|
| Vegetated wetland (*VW*) | % | 0 | 0.1 | 2.2 | 0.3 ± 0.4 |
| Urban (*UR*) | % | 0 | 0.1 | 5.8 | 0.4 ± 0.7 |
| Barren/Other (*BO*) | % | 0 | 1.3 | 30.0 | 4.4 ± 6.5 |
| **Land Disturbance Variables** | | | | | |
| Catchment disturbance ($D_C$) | % | 0 | 3.4 | 10.5 | 3.6 ± 2.1 |
| *HG* disturbance ($D_{HG}$) | % | 0 | 4.4 | 34.9 | 6.0 ± 6.4 |
| *PF* disturbance ($D_{PF}$) | % | 0 | 9.9 | 27.8 | 10.4 ± 6.7 |
| Stock unit density (*SUD*) | SU/ha | 0 | 2.2 | 16.1 | 3.2 ± 3.1 |
| Dairy SUD ($SUD_{da}$) | SU/ha | 0 | 0.2 | 15.4 | 1.2 ± 2.4 |
| Beef SUD ($SUD_{be}$) | SU/ha | 0 | 0.5 | 3.5 | 0.7 ± 0.8 |
| Sheep SUD ($SUD_{sh}$) | SU/ha | 0 | 0.6 | 4.5 | 1.2 ± 1.3 |
| Deer SUD ($SUD_{de}$) | SU/ha | 0 | 0 | 0.2 | 0 ± 0 |








Table 5. Statistical description of medians of water quality variables for the 77 NRWQN
catchments. Note that the ratio of mean/median can be used as an index of data skewness.

| Variable | Units | Minimum | Median | Maximum | Mean ± SD |
|---|---|---|---|---|---|
| $T_w$ | °C | 7.2 | 12.2 | 16.9 | 12.4 ± 2.4 |
| $DO$ | % | 75.5 | 100.8 | 113.1 | 100.0 ± 4.7 |
| $COND$ | µS/cm | 39 | 92 | 528 | 113 ± 83 |
| $pH_W$ | $-\log_{10}[H^+]$ | 6.9 | 7.7 | 8.5 | 7.7 ± 0.3 |
| $CLAR$ | m | 0.1 | 1.5 | 9.8 | 2.1 ± 1.8 |
| $TURB$ | NTU | 0.3 | 2.1 | 82 | 4.2 ± 9.4 |
| $CDOM$ | $m^{-1}$ | 0.1 | 0.7 | 4.6 | 0.9 ± 0.8 |
| $TN$ | $mg/m^3$ | 40 | 259 | 2162 | 369 ± 361 |
| $NO_x$ | $mg/m^3$ | 1 | 107 | 1852 | 230 ± 302 |
| $TP$ | $mg/m^3$ | 3 | 15 | 115 | 24 ± 24 |
| $DRP$ | $mg/m^3$ | 0.5 | 5.0 | 66.2 | 8.6 ± 11.2 |




Table 6. River water quality trends from 1989-2014. The table reports numbers of sites (out of 77) in different categories of water quality time trend. All variables were flow-adjusted except flow and water temperature. Significant trends were taken to be those with a p-value < 0.05 in the Seasonal Kendall test. Meaningful trends were taken to be those which also had a magnitude (RSKSE) greater than 1% per year.

| Direction of trend | River Water Quality Variable (1989-2014) | | | | | | | | | | | |
|---|---|---|---|---|---|---|---|---|---|---|---|---|
| | $Q$ | $T_w$ | DO | COND | $pH_w$ | CLAR | TURB | CDOM | TP | DRP | TN | $NO_x$ |
| Meaningful Increase | 1 | 0 | 0 | 4 | 0 | 29 | 17 | 1 | 8 | 17 | 27 | 24 |
| Significant Increase | 1 | 21 | 6 | 48 | 12 | 5 | 1 | 1 | 6 | 3 | 6 | 3 |
| No Significant Trend | 67 | 54 | 42 | 19 | 48 | 39 | 50 | 56 | 52 | 49 | 39 | 37 |
| Significant Decrease | 3 | 2 | 29 | 6 | 17 | 2 | 0 | 13 | 4 | 5 | 3 | 1 |
| Meaningful Decrease | 5 | 0 | 0 | 0 | 0 | 2 | 9 | 6 | 7 | 3 | 2 | 12 |





Table 7. Correlations of water quality (median values) vs. the major land uses, livestock densities, and median catchment disturbance of the 77 NRWQN catchments. All values represent Spearman correlation coefficients ($r_s$). Nonsignificant relationships ($p \geq 0.05$) are denoted by *NS*. $T_w$ was not included because of its strong latitudinal trend. *DO* and $pH_w$ were not included because they had no significant relationships with land use. $SUD_{cattle}$ is the combination of dairy and beef cattle.

| | *HG* | *SG* | *NF* | *PF* | *OW* | $SUD_{da}$ | $SUD_{be}$ | $SUD_{cattle}$ | $SUD_{sh}$ | $SUD_{de}$ | $D_C$ | $D_{HG}$ | $D_{PF}$ |
|---|---|---|---|---|---|---|---|---|---|---|---|---|---|
| *COND* | 0.57 | -0.53 | *NS* | 0.53 | *NS* | 0.44 | 0.63 | 0.60 | 0.35 | *NS* | *NS* | -0.25 | *NS* |
| *CLAR* | -0.45 | *NS* | 0.28 | -0.31 | *NS* | -0.41 | -0.49 | -0.49 | -0.40 | *NS* | *NS* | *NS* | -0.27 |
| *TURB* | 0.46 | *NS* | -0.27 | 0.28 | *NS* | 0.38 | 0.50 | 0.48 | 0.40 | *NS* | *NS* | *NS* | *NS* |
| *CDOM* | 0.56 | -0.55 | *NS* | 0.24 | -0.29 | 0.48 | 0.53 | 0.57 | 0.24 | *NS* | *NS* | -0.33 | *NS* |
| *TN* | 0.82 | -0.56 | -0.37 | 0.46 | -0.25 | 0.79 | 0.75 | 0.85 | 0.60 | 0.26 | *NS* | -0.40 | *NS* |
| $NO_x$ | 0.70 | -0.53 | -0.25 | 0.44 | -0.25 | 0.77 | 0.65 | 0.79 | 0.51 | 0.28 | *NS* | -0.39 | *NS* |
| *TP* | 0.66 | -0.54 | -0.32 | 0.48 | *NS* | 0.58 | 0.66 | 0.72 | 0.42 | *NS* | *NS* | -0.24 | *NS* |
| *DRP* | 0.59 | -0.65 | *NS* | 0.50 | -0.43 | 0.58 | 0.58 | 0.66 | 0.31 | *NS* | *NS* | -0.32 | *NS* |



Table 8. Stepwise regressions of water quality variables (median values) on landscape descriptors (forward selection, $p < 0.05$). Signs of coefficients indicate whether the relationship is proportional (+) or inverse (-). Int is model intercept.

| Water Quality Variable | Step | Landscape Variable | Model Estimate | Multivariate sequential $r^2$ |
|---|---|---|---|---|
| *CLAR* | 1 | *HG* | -0.03 | 0.17 |
| | 2 | *OW* | 0.18 | 0.27 |
| | 3 | $Q_{50}$ | -0.01 | 0.35 |
| | 4 | *PF* | -0.03 | 0.39 |
| | Int | | 3.16 | |
| *TN* | 1 | $SUD_{cattle}$ | 77.05 | 0.62 |
| | 2 | *HG* | 4.26 | 0.68 |
| | 3 | *PF* | 5.16 | 0.69 |
| | 4 | *SC%* | 1.80 | 0.72 |
| | Int | | -33.95 | |
| $NO_x$ | 1 | $SUD_{cattle}$ | 86.15 | 0.58 |
| | Int | | 62.65 | |
| *TP* | 1 | $SUD_{cattle}$ | 5.47 | 0.41 |
| | 2 | *PF* | 0.64 | 0.52 |
| | Int | | 7.75 | |
| *DRP* | 1 | $SUD_{cattle}$ | 2.23 | 0.31 |
| | 2 | *PF* | 0.38 | 0.48 |
| | Int | | 1.14 | |





**Figures**

**Figure 1**. Land use and location of the 77 National River Water Quality Network (NRWQN) catchments. Catchment ID colors refer to dominant land use (>50%). Catchments with no dominant land use are black.

**Figure 2**. Disturbance frequency of North Island per 463-m pixel, based on interpretation of MODIS data 2000-2013.

**Figure 3**. Disturbance frequency of South Island per 463-m pixel, based on interpretation of MODIS data 2000-2013.

**Figure 4**. Multivariate relationships between major water quality variables (median value for each site) and land use variables. For each plot, the primary explanatory variable from the stepwise regression (Table 8) is the x-axis, with bubble color representing the secondary explanatory variable. Note that oxidized nitrogen ($NO_x$) did not have a secondary explanatory variable. Selected catchments discussed in the text are labeled.

**Figure 5**. River water quality classes for upland (A) and lowland (B) catchments in New Zealand: I. clean river with high visual water clarity (CLAR) and low dissolved inorganic nutrients (DIN); II. sediment-impacted river with low CLAR and low DIN; III. nutrient-impacted river with high CLAR and high DIN; and IV. sediment- and nutrient-impacted river with low CLAR and high DIN. Classes are organized by ANZECC (2000) trigger values. DIN trigger values can be discriminated for $NO_x$ (y-axis) and $DRP$ (grey-filled markers). Arrows indicate whether the trend from 1989-2014 was significant (dashed) or meaningful (solid). No arrow means the trend was not significant.



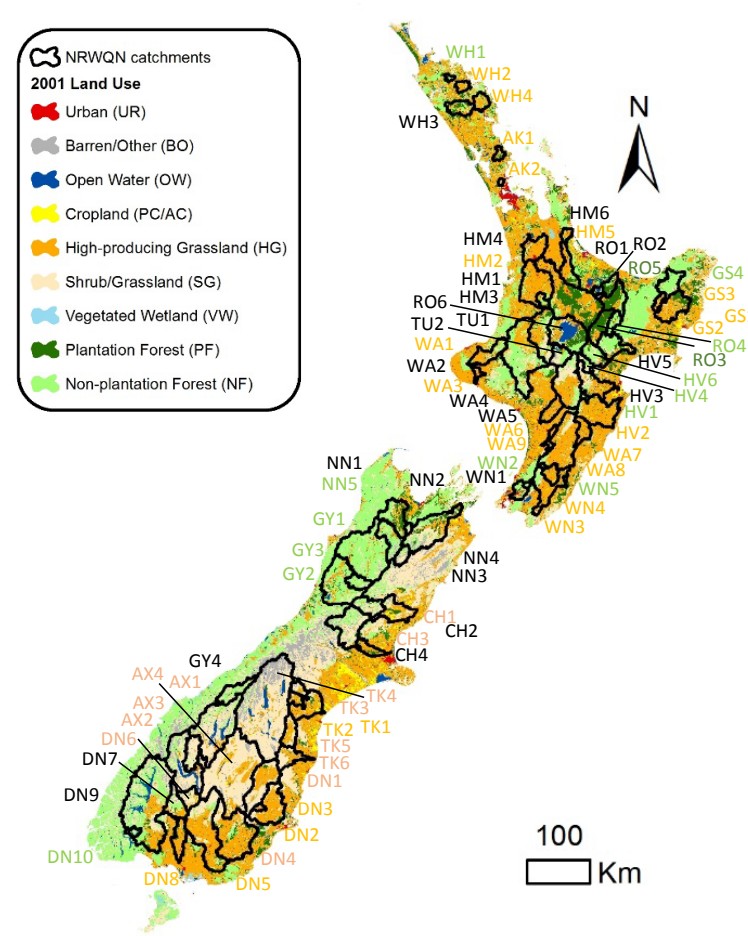

Figure 1. Land use and location of the 77 National River Water Quality Network (NRWQN) catchments. Catchment ID colors refer to dominant land use (>50%). Catchments with no dominant land use are black.

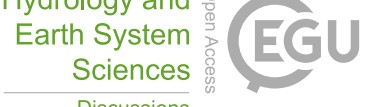



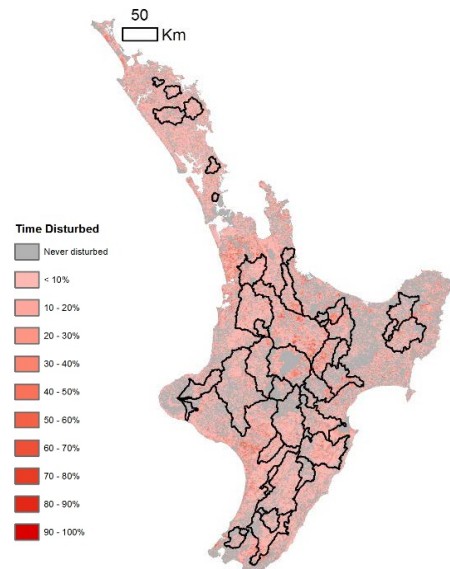

Figure 2. Disturbance frequency of North Island per 463-m pixel, based on MODIS data 2000-2013.



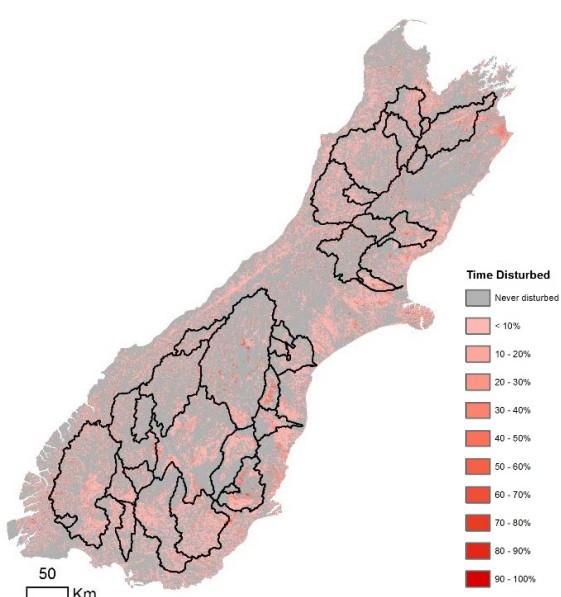

Figure 3. Disturbance frequency of South Island
per 463-m pixel, based on MODIS data 2000-
2013.





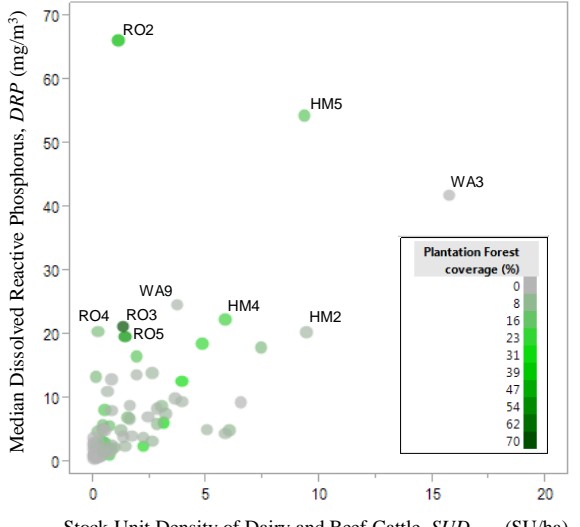

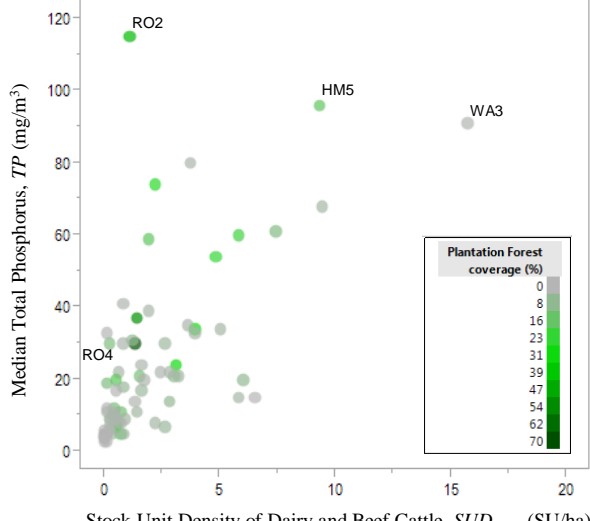

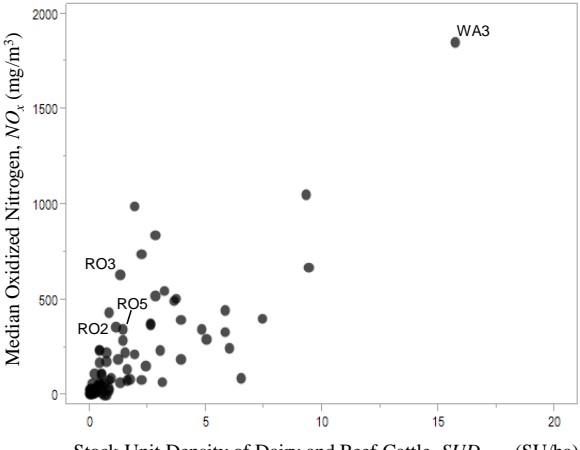

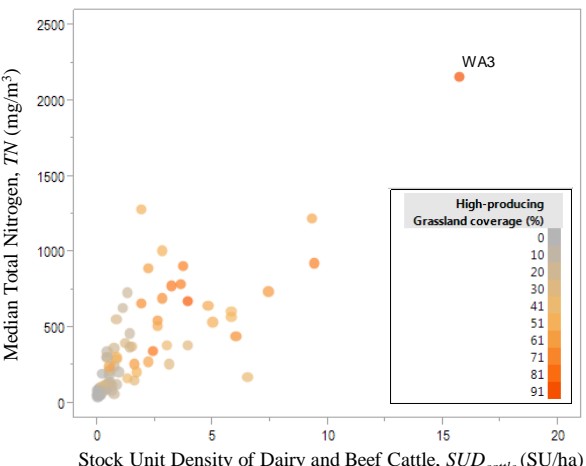

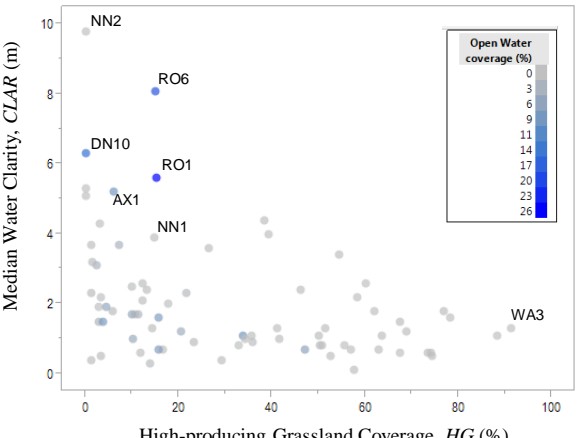

Figure 4. Multivariate relationships between major water quality variables (median value for each site) and land use variables. For each plot, the primary explanatory variable from the stepwise regression (Table 8) is the x-axis, with bubble color representing the secondary explanatory variable. Note that oxidized nitrogen ($NO_x$) did not have a secondary explanatory variable. Selected catchments discussed in the text are labeled.





## A. Upland Catchments

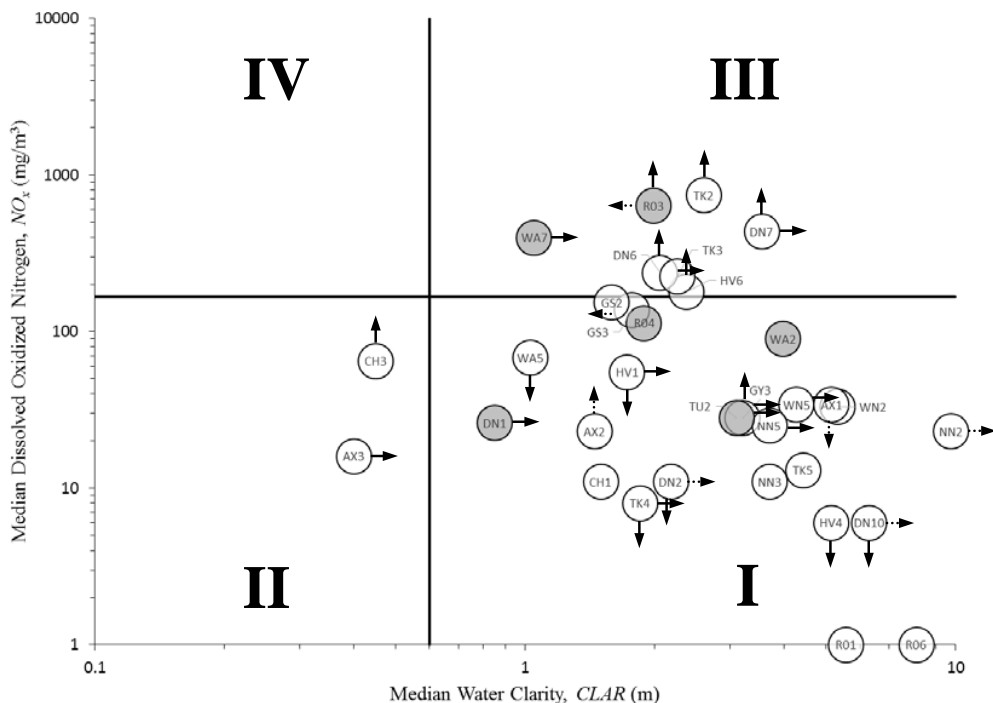

## B. Lowland Catchments

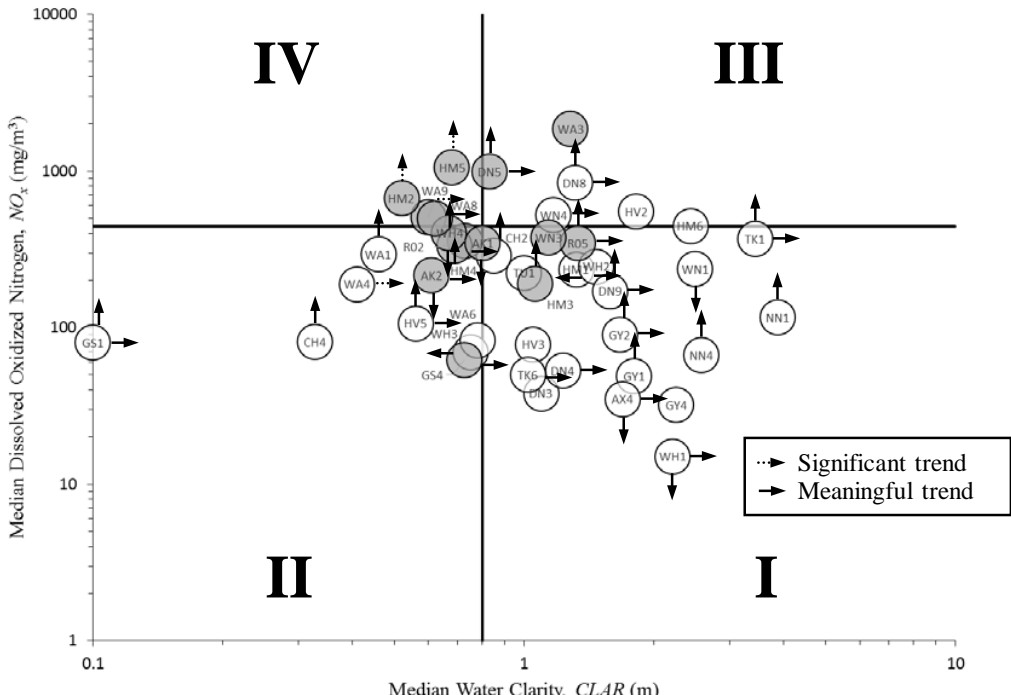

Figure 5. River water quality classes for upland (A) and lowland (B) catchments in New Zealand: I. clean river with high water clarity (CLAR) and low dissolved inorganic nutrients (DIN); II. sediment-impacted river with low CLAR and low DIN; III. nutrient-impacted river with high CLAR and high DIN; and IV. sediment- and nutrient-impacted river with low CLAR and high DIN. Classes are organized by ANZECC (2000) trigger values. DIN trigger values can be discriminated for $NO_x$ (y-axis) and $DRP$ (grey-filled markers). Arrows indicate whether the trend from 1989-2014 was significant (dashed) or meaningful (solid). No arrow means the trend was not significant.