# Peer review of "River water quality changes in New Zealand over 26 years (1989 – 2014): Response to land"

_Hydrology and Earth System Sciences, 2016_

## Referee Comment (RC1) · Anonymous Referee #1 · 21 Aug 2016

General comments The manuscript provides a comprehensive, robust and well-written analysis of relationships between landscape and land use indicators, and water quality in New Zealand. The research adds to analyses already conducted on water quality spatial and temporal trends in New Zealand. The manuscript is considered publishable in its current state, with some minor clarifications suggested below. I recommend retaining all the supplied supplementary material.

Specific comments L202 Were stock numbers normalized by total catchment area or by area of grassland per catchment? If the former, please comment on whether there were scenarios of under-estimation of stock density due to non-grassland area. L240 Were all the landscape variables independent of each other? L392 Did the sites that

had positive relationships between NOx and Q coincide with those which had positive relationships between P and Q? L436 Can some explanation for the 'Nineteen of the catchments experienced significant or 'meaningful' decreases in CDOM since 1989.' be provided? L770 I assume a reduction in N input to streams due to increased wetland coverage would be at a pollution swap cost of increased greenhouse gas emissions. The authors may wish to consider including a comment around this.

Technical comments L16 Can a more specific phrase than 'outweighs' be used? L16-17 check grammar – missing word been i.e. '...been cleaned up...' L21 Suggest reword 'from 1989 and 2014' to 'from 1989 to 2014' L 22 Suggest breaking in to two sentences: 'The NRWQN consisted of 77 sites...' L26 Please clarify if 'nitrate-nitrite-nitrogen' means both nitrate-N and nitrite-N, possibly by using the term 'total oxidised nitrogen'. L28 Suggest 'greatest negative impact' L34 Suggest rewords as ' and mobilized nutrients.' L35 Suggest reword as '..expertise in fields of geospatial analysis...' L45 and L48 Please provide appropriate references as these cause-effect relationships may not be ubiquitous L52 Re 'Historically, water quality in rivers was managed to meet minimal standards' In rivers where? And what is meant by minimal standards? L84 and throughout. Suggest avoiding abbreviations in the first word of a sentence L85 Suggest rewords as '...highest rates of agricultural land intensification...' L86-87 I suggest the NZ example would be of value to other developed countries that are intensifying, and possibly of less relevance to developing countries, where point source issues may still dominate water quality problems. Please consider. L123 suggest adding words for clarity '.....(NRWQN), which is operated and maintained by...' Table 1 An added column containing references to the methods used to analyse each parameter would be helpful. L147 Which DEM is referred to as 'This DEM...'? and why was rescaling from 25m to 30m resolution required? L150 Please describe the River Environment Classification (REC, v2.0) to the extent that your methods can be repeated by a reader. L153 Suggest italicising 'sensu' L154 Suggest reword as 'ratio of the total length of REC streams to catchment area.' L169 Please define 'annual water yield' L180 Please clarify the meaning of '...and maps land use for 2008 and 2012 as well for 12 classes.',

particularly re 12 classes of what? L191-192 Past tense if probably appropriate here. L210 Suggest reword as 'Land disturbance, defined here as bare soil, . . .' L212 Can you describe the extent to which you, or de Beurs et al (2016) ground-truthed the disturbance index against measures of bare soil, to give the reader confidence in the method? L231-232 Please clarify what is meant by 'actual values' in this context. L371 and 372 Please clarify – COND and pH decreased with increasing Q? L396, 451 Check spelling of phosphorus L513. Which four nutrient variables are referred to here? Figure 5 What does the direction of the symbol arrows mean? Also, I suggest using the explanation in the text 'catchments that exceed ANZECC guidelines for DRP are indicated in by grey-filled markers' rather than the current caption text because the latter is hard to understand.

―――――――――――――――

---

## Referee Comment (RC2) · Anonymous Referee #2 · 26 Aug 2016

General comment

The manuscript "River water quality changes in New Zealand over 26 years (1989-2014): Response to land use and land disturbance" aims at investigating the relationships between land use and water quality, means suspended sediment, nitrogen and phosphorus concentration. A huge database of monthly data from 26 years and 77 watersheds has been analyzed. The main concern I have with the manuscript is that based on relatively simple statistical analysis mostly only weak correlation could be achieved between land use and selected water quality variables. Nearly all presented Spearman rank correlation coefficients were relatively low (approximately below 0.7 or even much lower). Choosing this correlation coefficient means that outcomes are descriptive and no quantitative functional dependences can be derived. Furthermore the study primarily focusses on suspended sediment driven water quality constituents like suspended sediment concentration, total nitrogen and total phosphorus but the analysis is restricted to monthly data, hence the most important short term events with high concentrations of the abovementioned compounds are not considered in the study. In general the manuscript is highly descriptive, clear hypotheses are missing, a sound reasoning for the approach based on former studies is not given, and some conclusions are made without clear evidence. Therefore it is not clear what is new and what has already been investigated in former studies. Furthermore the manuscript is very long (41 pages text only) and not very specific including repetitions. Therefore I reject the manuscript and suggest a submission to a journal with a more narrow scope.

Specific comments

Abstract: Not very specific, no specific numerical results are given on land use impacts on SSC and nutrient concentrations

Introduction: The first section is too general and not enough focused on the study presented. Review of international literature on dependencies between land use and water quality is weak. In Europe and the US much effort has been undertaken e.g. in connection with the SPARROW model (regression based water quality model) or time series analyses on land use induced N and C losses ( e.g. in the UK; Worrall et al.). A focus could be given on grassland impacts. Some former studies from New Zealand are mentioned, e.g. time series analysis of nearly the same data set (see References), but the manuscript could benefit much more from these studies if they would be used to clearly identify the present knowledge and use them to define research questions which are still open. The objectives are not quite clear. The study wants to illustrate long-term relationships among land management, geomorphic processes and river water quality. It is not clear why temporal changes in land use should be compared to temporal changes in water quality variables (line 92) when the investigation later on shows that land use changes are usually minor (line 313) or mostly negligible.

[Figure]

Line 281: These are very general and obvious findings, e.g. that discharge increases with catchment area, I would take this out

Line 289-312: Land use distribution and patterns should be regarded as site description and not presented as part of the results (the objective of the study is to investigate relationships between Water quality and land use)

Line 313: The same is true for the land use change description, especially because land use change was mostly negligible.

Line 324: livestock densities expressed as SU?

Line 349: Relationships between disturbance (by the way a clear definition of disturbances would be helpful) and catchment characteristics should be restricted to those which are a) significant and pronounced and b) meaningful, for example for me it is unclear why disturbances has been related to mean annual sunshine duration, what does a rs of -0.25 tell us?

Line 368ff: Very general outcomes are presented which we would expect in any catchment, e.g. that suspended sediment concentration decrease with flow, furthermore most of the findings have already been reported elsewhere (20 years ago, see references of the manuscript) for the study region. Because of the suggested statistical analysis no detailed functional quantitative relationships can be defined.

Line 404: This section is very descriptive and simply repeats what is shown in Table 5 and table 6. A restriction on significant trends would be useful.

Line 439: The statement that total nitrogen was high if the concentrations are above 0.25 mgN/l is at least questionable. I would asses these levels still as pristine. The same is true for the assessment of TP with concentrations of 0.03 mg/l and DRP of 0.009 mg/l assumed as high levels. In Europe the eutrophication level for DRP is 0.05 for streams. Please revise.

Line 459: It is boring to be informed again that discharge increases with catchment

area. Please take this out, a number in a table should be sufficient

Line 469: This correlation is quite small and should not be over interpreted, is there a table available on these correlations?

Line 491ff: In the result section wordings like surprisingly , interestingly should be obmitted, especially if an additional discussion section follows. By the way why is it interesting that TP is increasing only in two catchments?

Line 542: What does this mean, only single values or 90% percentiles?

Line 545: Please clearly state in the introduction which work has already been done regarding to state and trend analysis (e.g. Ballantine and Davies-Colley Water quality trends in New Zealand rivers: 1989-2009 Environmental Monitoring and Assessment 186, 3, 1939-1950) Line 617: What does this tell us, recovery is quick because of high Pret? But high Pret should decrease the release of P from soil to soil pore water? Line 667: 0.45 is definitely not a high correlation, please correct Line 672: already discussed before Line 896: it is assumed that there is no doubt that fencing has improved the clarity of NZ rivers, but this is only a possible explanation. No evidence is given in the manuscript. In the presented study no specific quantitative analysis on fencing and sediment concentrations has been conducted.

Reference:

Ballantine DJ, and Davies-Colley, RJ, 2014, Water quality trends in New Zealand rivers: 1989-2009 Environmental Monitoring and Assessment 186, 3, 1939-1950 Worrall, F., H. Davies, T. Burt, N. J. K. Howden, M. J. Whelan, A. Bhogal, and A. Lilly. 2012. The flux of dissolved nitrogen from the UK - Evaluating the role of soils and land use. Science of the Total Environment 434:90-100.

---

## Editor Comment (EC1) · C. Stamm (Editor) · 29 Aug 2016

Dear Dr. Julian

Below, I listed some comments that might be relevant for the interactive discussion of this manuscript.

General comment:

**Novelty of the results**: It is difficult to identify what is novel with this manuscript as compared to other published studies on water quality trends in New Zealand rivers (and in general). In the Introduction for example, the reader cannot find clear statements about findings from the NZ monitoring activities and the open question that need to be addressed (e.g., L. 79 – 82: what are the reported trends, what are open questions?). In which sense does this manuscript add to previously published work? At the same time, there is a lack of international context that illustrates which scientific questions should be tackled and how this study contributes to generating new insight.

Specific comments:

L. 20: The term "disturbance" is not clear for a reader at that stage of reading. Only when going through all your later explanations it gets obvious what is meant by this term.

L. 34 – 37: This conclusion is not really based on results in this manuscript but refers to an interdisciplinary approach that is quite common in studies on land use and water quality.

L. 40: This statement does hold in its generality – many important human activities (e.g., arts, science etc.) are not reflected in water quality.

L. 55 – 64: These sentences evoke the impression that mitigation has only focused on point-source pollution and that causal understanding about diffuse pollution is lacking. However, abating diffuse pollution has been pursued in many countries for several decades and many essentials are known about the drivers of diffuse pollution. Corresponding monitoring activities have been implemented and yield insight about success and short-comings of such programs (see (Kronvang *et al.* 2008) as just one arbitrary example). Your wording should better reflect the international state-of-the-art.

L. 65 – 67: I think this statement does not hold true (see comment above).

L. 73 – 74; 87 – 88; 124 - 125: I doubt whether these statement hold true: There are many countries that have well-designed monitoring programs that run for several decades already. Some of them may also have more advanced sampling strategies than what you describe in this manuscript. In Switzerland for example, the national monitoring program starting in the 1970s for some parameters provides decade-long time-series of flow-proportional samples at weekly or bi-weekly resolution instead of monthly grab samples (Zobrist & Reichert 2006; Zobrist 2010). In the UK, some time-series date back in time over 100 yrs and make clear links between water quality and land use (Howden *et al.* 2011). Your statements should better reflect what others have been doing for quite some time.

L. 161: P retention: please provide a reference for this method.

L. 223 – 228: Do you have ground truth data?

Table 1: Give the temporal resolution of the data sets.

Table 8: For judging the quality of the regressions one should see scatter plots of the actual data.

Fig. 4: The plots reveal  pronounced heteroscedasticity. How did you deal with this issue?

Sincerely

Dr. Christian Stamm
Editor HESS
Environmental Chemistry
Eawag
Ueberlandstr. 133
CH-8600 Dübendorf, Switzerland

References:

Howden, N.J.K., Burt, T.P., Worrall, F., Whelan, M.J. & Bieroza, M. (2011) Nitrate concentrations and fluxes in the River Thames over 140 years (1868–2008): are increases irreversible? *Hydrological Processes,* **24,** 2657-2662.

Kronvang, B., Andersen, H.E., Børgesen, C., Dalgaard, T., Larsen, S.E., Bøgestrand, J. & Blicher-Mathiasen, G. (2008) Effects of policy measures implemented in Denmark on nitrogen pollution of the aquatic environment. *Environmental Science & Policy,* **11,** 144 - 152.

Zobrist, J. (2010) Water Chemistry of Swiss Alpine Rivers. *Alpine waters* (ed. U. Bundi), pp. 95 - 118. Springer.

Zobrist, J. & Reichert, P. (2006) Bayesian estimation of export coefficients from diffuse and point sources in Swiss watersheds. *Journal of Hydrology,* **329,** 207–223; http://dx.doi.org/210.1016/j.jhydrol.2006.1002.1014.

---

## Author Comment (AC1) · 14 Oct 2016

**Authors' Reply to Referee #1**

**Major issues:**

none

**Specific comments:**
L202 Were stock numbers normalized by total catchment area or by area of grassland per catchment? If the former, please comment on whether there were scenarios of under-estimation of stock density due to non-grassland area.
We added the identifier 'total' before catchment area to avoid confusion. We addressed the 2nd issue in section 5.2 where we acknowledge that "Our catchment-scale analyses limit our interpretation of specific situations, …" In this section, we also identify physiographic locations that will likely have higher livestock densities. If desired, we can add a few sentences in the Results that report maximum livestock densities at the 1 ha pixel scale.

L240 Were all the landscape variables independent of each other?
No. We report the correlations among landscape variables in section 4.1. At the end of section 4.1, we state that: "we used the following subset of minimally correlated physiographic variables for subsequent multivariate analyses: catchment slope ($S_c$), silt-clay percentage ($SC\%$), phosphate retention ($P_{ret}$), and median flow ($Q_{50}$)."

L392 Did the sites that had positive relationships between NOx and Q coincide with those which had positive relationships between P and Q?
In terms of TP v Q, yes because almost all sites (73/77) displayed a positive relationship between TP and Q. In terms of DRP, all but 3 displayed the same pattern. We added this relationship: 'Dissolved reactive phosphorus (*DRP*) generally increased with $Q$ for most sites, including all the same sites that displayed a positive relationship between $NO_x$ and $Q$ except three (GS2, HM1, HM2).'

L436 Can some explanation for the 'Nineteen of the catchments experienced significant or 'meaningful' decreases in CDOM since 1989.' be provided?
We added: "…,possibly due to the loss of wetlands across NZ."

L770 I assume a reduction in N input to streams due to increased wetland coverage would be at a pollution swap cost of increased greenhouse gas emissions. The authors may wish to consider including a comment around this.
The answer to this question is not straightforward. For wetland restoration, greenhouse gas emissions from wetlands (methane mainly, not much nitrous oxide although we don't have enough knowledge) tend to be compensated by carbon sequestration, thus in the long-term freshwater wetlands may be better than 'carbon-neutral'. By restoring wetlands with forested cover, a sink for GHGs might be provided. We also think that the area restored would be too small for GHG emissions to be an issue (*note the C sequestration from potential added trees). If the areas restored were to be substantial, we should then also consider that that area could be

converted FROM pastoral areas where animals contributing to GHG emissions could be removed (thus compensating out). A good reference for this is:

*Bridgham, S.D., Moore, T.R., Richardson, C.J. et al. Landscape Ecol (2014) 29: 1481. doi:10.1007/s10980-014-0067-2*

Note their conclusions: "Because wetlands provide many ecosystem services in addition to C sequestration, it is short-sighted to suggest that wetlands should not be created or restored because of their [GHG] emissions." We can discuss this in our paper, but as you see, to do justice to this complex issue would require a further paragraph in an already long ms.

**Technical comments:**

Note: We have rewritten the Abstract (see response to Editor), so abstract comments may no longer apply.

L16 Can a more specific phrase than 'outweighs' be used?
Changed to: has a more extensive and appreciable impact

L16-17 check grammar – missing word been i.e. ': : :been cleaned up: : :'
Corrected as suggested.

L21 Suggest reword 'from 1989 and 2014' to 'from 1989 to 2014'
Corrected as suggested.

L 22 Suggest breaking in to two sentences: 'The NRWQN consisted of 77 sites: : :'
Changed as suggested.

L26 Please clarify if 'nitrate-nitrite nitrogen' means both nitrate-N and nitrite-N, possibly by using the term 'total oxidized nitrogen'.
Changed to: dissolved oxidized nitrogen

L28 Suggest 'greatest negative impact'
Changed as suggested.

L34 Suggest rewords as ' and mobilized nutrients.'
Reworded as suggested.

L35 Suggest reword as '..expertise in fields of geospatial analysis: : :'
Reworded as suggested.

L45 and L48 Please provide appropriate references as these cause-effect relationships may not be ubiquitous
We added the McDowell et al. (2008) reference which documents these effects for rivers in general and specifically for New Zealand.

L52 Re 'Historically, water quality in rivers was managed to meet minimal standards' In rivers where? And what is meant by minimal standards?

This is a statement about global rivers in general. We realize that the reference we used was for U.S. rivers, so we added another, more global reference (Boesch, 2002) and one specific to NZ (Howard-Williams et al., 2010). We also revised the sentence to say 'minimally acceptable standards or maximum pollutant load limits.'

L84 and throughout. Suggest avoiding abbreviations in the first word of a sentence
We corrected this instance and will correct other instances in the revision.

L85 Suggest rewords as ': : :highest rates of agricultural land intensification: : :'
Reworded as suggested.

L86-87 I suggest the NZ example would be of value to other developed countries that are intensifying, and possibly of less relevance to developing countries, where point source issues may still dominate water quality problems. Please consider.
Good point. We believe it is relevant to both developed and developing countries. Thus, we removed 'some developing.'

L123 suggest adding words for clarity ': : :..(NRWQN), which is operated and maintained by: : :'
Added as suggested.

Table 1 An added column containing references to the methods used to analyse each parameter would be helpful.
Analytical methods for all variables can be found in one reference: Davies-Colley et al. (2011). We added to the Table 1 caption: Details on analytical methods can be found in Davies-Colley et al. (2011). We can add an extra column detailing each method, but it would make this table much larger, equivalent to one full page.

L147 Which DEM is referred to as 'This DEM: : :'? and why was rescaling from 25m to 30m resolution required?
We added that we rescaled 'in order to align with other gridded spatial datasets.' We also clarified in the next sentence that 'This 25-m DEM …'

L150 Please describe the River Environment Classification (REC, v2.0) to the extent that your methods can be repeated by a reader.
This is a publically available dataset, described by Snelder et al. (2010). We added: "…,the national hydrography dataset derived from a 30-m hydrologically correct DEM (Snelder et al., 2010)."

L153 Suggest italicising 'sensu'
Italicized.

L154 Suggest reword as 'ratio of the total length of REC streams to catchment area..'
Reworded as suggested.

L169 Please define 'annual water yield'
We clarified that this is: total volume of water leaving the catchment in a year.

L180 Please clarify the meaning of ': : :and maps land use for 2008 and 2012 as well for 12 classes.', particularly re 12 classes of what?
We revised the sentence as follows: "Accordingly, LUCAS uses 1990 as its reference year and maps land use in 12 classes for 2008 and 2012."

L191-192 Past tense if probably appropriate here.
Both sentences converted to past tense.

L210 Suggest reword as 'Land disturbance, defined here as bare soil, : : :'
Revised as suggested, but added "Land disturbance, defined here as bare soil resulting from vegetation loss, …".

L212 Can you describe the extent to which you, or de Beurs et al (2016) ground-truthed the disturbance index against measures of bare soil, to give the reader confidence in the method?
We added the following to the end of section 3.3: MODIS disturbance data were visually validated against 7500 random pixels from Landsat imagery and corresponding 15 high resolution Orbview-3 and Ikonos images. The overall accuracy of the disturbance index based on Landsat data was 98%.

L231-232 Please clarify what is meant by 'actual values' in this context.
We meant correlation parameters (Pearson's r values), but recognize this as written, was confusing. We have removed this term to avoid confusion.

L371 and 372 Please clarify – COND and pH decreased with increasing Q?
The word 'increasing' was added to both sentences for clarity.

L396, 451 Check spelling of phosphorus
Corrected both of these, and L445 as well.

L513. Which four nutrient variables are referred to here?
The four nutrient variables in Table 8, which is referenced at the end of the sentence. For clarity, we added: "Dissolved oxidized nitrogen ($NO_x$) was not proportional to $PF$, or any other independent variable in the stepwise regression."

Figure 5 What does the direction of the symbol arrows mean? Also, I suggest using the explanation in the text 'catchments that exceed ANZECC guidelines for DRP are indicated in by grey-filled markers' rather than the current caption text because the latter is hard to understand.
Figure caption revised as suggested. Explanation of arrow direction was also added: "Arrows indicate direction of trend over the 26 years inclusive from 1989 if significant (dashed) or meaningful (solid). No arrow means the trend was not significant."

---

## Author Comment (AC2) · 14 Oct 2016

**Authors' Reply to Referee #2**

**Major issues:**

1. The main concern I have with the manuscript is that based on relatively simple statistical analysis mostly only weak correlation could be achieved between land use and selected water quality variables. Nearly all presented Spearman rank correlation coefficients were relatively low (approximately below 0.7 or even much lower). Choosing this correlation coefficient means that outcomes are descriptive and no quantitative functional dependences can be derived.

We used relatively simple statistical analyses for the spatial comparisons because the patterns were obvious. Also, we believe the simplest explanations are more convincing to a broader audience (Occam's razor). For the temporal statistical analyses, we used the seasonal Mann-Kendall test, which is an appropriate test of temporal trends of monthly data. Many of the temporal trends we found were highly significant, with many also being 'meaningful' (i.e. greater than 10% change per decade). As for the relatively low correlation, these are large catchments with a considerable amount of physiographic variability, interacting factors, and feedbacks. When relationships are developed for these 77 large, diverse catchments, it is inevitable that correlations tend to be weaker than for small homogenous catchments or longitudinally along the same catchment.

2. Furthermore the study primarily focusses on suspended sediment driven water quality constituents like suspended sediment concentration, total nitrogen and total phosphorus but the analysis is restricted to monthly data, hence the most important short term events with high concentrations of the abovementioned compounds are not considered in the study.

Our study addresses both particulate and dissolved constituents. For each site, a wide range of flows were sampled over the 312 visits, including high flow events (range for all sites: $0 – 4,230$ m$^3$/s). Similarly, TN ranged $5 – 7,000$ mg/m$^3$ and TP ranged $0 – 8,000$ mg/m$^3$ among all sites. If desired, we can include a supplementary table with the ranges of discharge and water quality variables for each site. It should be noted, however, that our spatial analyses among the 77 catchments assessed the median condition of water quality variables. Furthermore, although targeted sampling of high flow events is very relevant for load estimation of particle-related contaminants, it is not appropriate for state-of-environment monitoring like NRWQN (e.g. Davies-Colley et al. 2011; cited), covering both dissolved and particulate constituents, for which random or pseudo-random (e.g. regular monthly as in the NRWQN) sampling is most appropriate.

3. In general the manuscript is highly descriptive, clear hypotheses are missing, a sound reasoning for the approach based on former studies is not given, and some conclusions are made without clear evidence. Therefore it is not clear what is new and what has already been investigated in former studies.

We realize that the novelty and contribution of our work was not made clear in the original Introduction. Accordingly, we have revised the Introduction considerably and now address the objections of this reviewer as regards: descriptive statistics, hypotheses, justification of the approach, novelty of our contribution and conclusions. We have also changed the title of the paper to: **River water quality changes in New Zealand over 26 years (1989 – 2014):**

**Response to land use intensity**. We have pasted the last four paragraphs of the new Introduction below, which lays out what has been done in terms of multi-catchment land-water relationships, our current lack of understanding of land use intensity, and our novel contribution:

[revised manuscript text omitted]

4. Furthermore the manuscript is very long (41 pages text only) and not very specific including repetitions.

The manuscript is long because of our comprehensive coverage of both spatial and temporal effects of land use on a wide range of river water quality variables in complex large catchments. Arguably, the paper could be split into two manuscripts, but we feel it will have a greater impact as one paper. Further, an understanding of temporal effects is necessary in order to explain some of the spatial effects, and vice versa. We do not understand the comment 'not very specific.' We did a lot of investigation on land use practices and processes that were responsible for the patterns and relationships we observed. Maybe the reviewer is referring to our scale of analysis: catchment-scale. On line 95, we state: "Most of our analyses were performed at the catchment scale because it integrates the spatiotemporal changes that are reflected in our water quality measurements, it is the appropriate scale to analyze diffuse pollution, and it is the most appropriate spatial management unit (Howard-Williams et al., 2010)."

**Specific comments:**
Abstract: Not very specific, no specific numerical results are given on land use impacts on SSC and nutrient concentrations

As our ms shows, land use does not uniquely (predictably) mobilize SSC and nutrients. It is the *interaction* of land use with geology/climate/physiography that is important, and that is why our paper lacks specific numerical results for these contaminants. We have rewritten the abstract (below) to highlight the novelty and contribution of our work, and to address several comments among the reviewers.

Land use-water quality relationships are complex with interdependencies, feedbacks, and legacy effects. Most river water quality studies have assessed catchment land use as areal coverage, but here, we hypothesize and test whether land use *intensity* – the inputs (e.g. fertilizer, livestock) and activities (e.g. vegetation removal) of land use – is a better predictor of environmental impact. We use New Zealand as a case study because it has had one of the highest rates of agricultural land intensification globally over recent decades and it has a long, consistent, and comprehensive national water quality dataset. We interpreted water quality state and trends for the 26 years from 1989 to 2014 in the National Rivers Water Quality Network (NRWQN) – consisting of 77 sites on 35 mostly large river systems with an aggregate catchment amounting to half of NZ's land area. To characterize land use intensity, we analyzed spatial and temporal changes in livestock density and land disturbance (i.e. bare soil resulting from vegetation loss by either grazing or forest harvesting) at the catchment-scale, as well as fertilizer inputs at the national scale. Using simple multivariate statistical analyses across the 77 catchments, we found that visual water clarity was best predicted by areal coverage of high-producing pastures. The primary predictor for all four nutrient variables, however, was cattle density, with plantation

forest coverage as the secondary predictor variable. While land disturbance was not itself a strong predictor of water quality, it did help explain outliers of land use-water quality relationships. From 1990 to 2014, visual clarity significantly improved in 34/77 catchments, which we attribute mainly to increased dairy cattle exclusion from rivers (despite dairy expansion) and the considerable decrease in sheep numbers across the NZ landscape, from 58 million sheep in 1990 to 31 million in 2012. Nutrient concentrations increased in many of NZ's rivers with dissolved oxidized nitrogen significantly increasing in 27/77 catchments, which we largely attribute to increased cattle density and legacy nutrients that have built up on high-producing grasslands and plantation forests since the 1950s and are slowly leaking to the rivers. Despite recent improvements in water quality for some NZ rivers, these legacy nutrients and continued agricultural intensification are expected to pose broad-scale environmental problems for decades to come.

Introduction: The first section is too general and not enough focused on the study presented. Review of international literature on dependencies between land use and water quality is weak. In Europe and the US much effort has been undertaken e.g. in connection with the SPARROW model (regression based water quality model) or time series analyses on land use induced N and C losses ( e.g. in the UK; Worrall et al.). A focus could be given on grassland impacts. Some former studies from New Zealand are mentioned, e.g. time series analysis of nearly the same data set (see References), but the manuscript could benefit much more from these studies if they would be used to clearly identify the present knowledge and use them to define research questions which are still open. The objectives are not quite clear. The study wants to illustrate long-term relationships among land management, geomorphic processes and river water quality. It is not clear why temporal changes in land use should be compared to temporal changes in water quality variables (line 92) when the investigation later on shows that land use changes are usually minor (line 313) or mostly negligible.

As we addressed in your 3rd Major Issue, we have revised the Introduction considerably and added several references for more international context. We did not talk about SPARROW studies because they address longitudinal/downstream changes in water quality, whereas we focus on differences among diverse catchments. The land use change we refer to on line 313 is areal coverage. One of the key and novel findings of our study is that while areal coverage has not changed, land use intensity (in the form of livestock densities and land disturbance) has changed; which explains some of the changes in water quality we observed. We have better articulated the difference between land use areal coverage and land use intensity in the Introduction and throughout the revised manuscript.

Line 281: These are very general and obvious findings, e.g. that discharge increases with catchment area, I would take this out
We agree that this is a general finding, but we need to include the correlation coefficient here in order to justify our selection of minimally correlated variables at the end of the paragraph.

Line 289-312: Land use distribution and patterns should be regarded as site description and not presented as part of the results (the objective of the study is to investigate relationships between Water quality and land use)
These distributions and patterns (and comparisons with physiography) resulted from our analyses of land cover and land use within and among the 77 NRWQN catchments. We would like to note

that our study is the most comprehensive analysis of land cover and land use change for the 77 NRWQN catchments. While previous studies have attributed changes in water quality from the NRWQN dataset to land use change, their comparisons (e.g. Ballantine and Davies-Colley, 2014) were not as rigorous as ours.

Line 313: The same is true for the land use change description, especially because land use change was mostly negligible.

The land use change we refer to on line 313 is areal coverage. One of the key and novel findings of our study is that while areal coverage has not changed much, land use *intensity* in the form of livestock densities and land disturbance has changed; which explains some of the changes in water quality we observed. We believe that is a novel and important contribution of our ms, and indeed we now highlight the focus on land use intensity in the revised title.

Line 324: livestock densities expressed as SU?

We report livestock densities in stock units per hectare (SU/ha), which is defined in section 3.3 (line 200).

Line 349: Relationships between disturbance (by the way a clear definition of disturbances would be helpful) and catchment characteristics should be restricted to those which are a) significant and pronounced and b) meaningful, for example for me it is unclear why disturbances has been related to mean annual sunshine duration, what does a rs of -0.25 tell us?

Previously, land disturbance was defined on line 210 as bare soil. In our rewritten abstract and throughout the revised manuscript, we provide clear definitions of land disturbance: "bare soil resulting from vegetation loss by either grazing or forest harvesting." As for the relatively low correlation, again, these are large catchments with a considerable amount of physiographic variability, interacting factors, and feedbacks, such that relationships are complex. We discuss the importance of sunshine duration on line 608: "Year-round and intense grazing is best supported by warm and sunny climates where pasture grasses are highly productive and recover quickly following intense grazing such as strip/rotational grazing which is common in NZ dairy farms." On lines 336, we also report that the highest cattle densities were found in sunny regions. Therefore for high-producing pastures with high sunshine hours, grasses recover quicker and thus have less disturbance over time.

Line 368ff: Very general outcomes are presented which we would expect in any catchment, e.g. that suspended sediment concentration decrease with flow, furthermore most of the findings have already been reported elsewhere (20 years ago, see references of the manuscript) for the study region. Because of the suggested statistical analysis no detailed functional quantitative relationships can be defined.

We agree that some of outcomes are general and intuitive; however, where we differ from previous studies on water quality in NZ is that we emphasize the *exceptions* including (but not limited to) effect of reservoirs on discharge-water quality relationships. By pointing out the exceptions, future studies can investigate their causes. Also, we believe it is a valuable contribution to provide empirical data over an entire country over 26 years to support these relationships. Further, some *HESS* readers may not be familiar with discharge-water quality relationships, and thus it is important to provide this foundation before moving on the land use effects on water quality. Finally, as already mentioned, in the revised ms we emphasize effects

on water quality of change in *intensity* of land use rather than areal coverage, and this is a novel and valuable contribution.

Line 404: This section is very descriptive and simply repeats what is shown in Table 5 and table 6. A restriction on significant trends would be useful.
This section reports the results from the temporal trend analyses, which is a key element of our paper. While Tables 5 & 6 are summaries of the data, section 4.3.2 provides details on the processes and spatial patterns of these trends, another key element of our paper. Throughout the paper, we only report significant relationships (alpha < 0.05).

Line 439: The statement that total nitrogen was high if the concentrations are above 0.25 mgN/l is at least questionable. I would asses these levels still as pristine. The same is true for the assessment of TP with concentrations of 0.03 mg/l and DRP of 0.009 mg/l assumed as high levels. In Europe the eutrophication level for DRP is 0.05 for streams. Please revise.
We have revised this section, using ANZECC (2000) guidelines for New Zealand to define *relatively* high vs. acceptable values - to recognize that, in NZ, nutrient levels in near-pristine catchments are very low. We added the word 'relatively' in front of the word high for this section. For TN, we changed the guideline to 455 mg/m$^3$, which is the midway point between trigger values for upland and lowland rivers. These same guidelines for TP and DRP are 30 and 9.5 mg/m$^3$, respectively.

Line 459: It is boring to be informed again that discharge increases with catchment area. Please take this out, a number in a table should be sufficient
We removed this sentence.

Line 469: This correlation is quite small and should not be over interpreted, is there a table available on these correlations?
No, currently there is no table for all the correlations. Such a table is very large given the 25 independent variables and 11 dependent variables, and we feel it would clutter an already long ms. We only report the 'significant' (alpha < 0.05) relationships because the many weaker ('insignificant') correlations are mostly of minor interest.

Line 491ff: In the result section wordings like surprisingly , interestingly should be obmitted, especially if an additional discussion section follows. By the way why is it interesting that TP is increasing only in two catchments?
We use these terms to note when there are novel or counterintuitive relationships, as is clear from the context. We then consider why these 'unexpected' relationships may have occurred. It is interesting that TP increased in only two of the highly disturbed catchments because we usually connect intensive land use and land disturbance (i.e. bare soil) with sediment runoff, which has phosphorus associated with it.

Line 542: What does this mean, only single values or 90% percentiles?
These are ANZECC guidelines for lowland/upland, respectively. We added 'ANZECC' in front of guidelines for clarity.

Line 545: Please clearly state in the introduction which work has already been done regarding to state and trend analysis (e.g. Ballantine and Davies-Colley Water quality trends in New Zealand rivers: 1989-2009 Environmental Monitoring and Assessment 186, 3, 1939-1950)

We did state this on lines 79-82: "Like Finland, New Zealand (NZ) has an extensive river water quality monitoring network, which has allowed many studies on river water quality state and trends (Smith et al., 1996, 1997; Scarsbrook et al., 2003; Scarsbrook, 2006; Ballantine and Davies-Colley, 2014) and effects of land use (Davies-Colley, 2013; Larned et al., 2004, 2016)." However, these earlier studies did not relate water quality to catchment land use intensity and land disturbance as we do in this ms. We now highlight the novel contributions of our paper.

Line 617: What does this tell us, recovery is quick because of high Pret? But high Pret should decrease the release of P from soil to soil pore water?

In this sentence, we state that these pastures with high $P_{ret}$ are managed intensively. In the three preceding sentences, we lay out the reasoning that they recover quickly because they receive large amounts of P-fertilizer, respond favorably to P-fertilizer, and are managed intensively.

Line 667: 0.45 is definitely not a high correlation, please correct

We changed it to 'relatively high correlation.'. Obviously correlation strength is context-dependent, and as previously-mentioned, high correlations cannot be expected for large, complex (e.g. multiple land-use) catchments.

Line 672: already discussed before

The first mention of riparian fencing (line 579) was related to general improvements in water clarity among all catchments, with references to previous studies that also attributed improvements to riparian fencing. In this second instance, we relate these improvements specifically to dairy cattle and the 2003 Dairying and Clean Streams Accord (requiring exclusion of dairy cattle from channels), in view of the known degrading effects of cattle on streambanks (Trimble and Mendel 1995). We have revised this 2nd mention of riparian fencing as follows: "Second, we found that *CLAR* has actually *improved* in catchments where *SUD$_{cattle}$* is high and/or has increased (Fig. 5), which we noted earlier could be a result of increased riparian fencing. In 2003, NZ implemented the *Dairying and Clean Streams Accord*, which has led to the exclusion of dairy cattle from 87% (as of 2012) of perennial rivers greater than 1 m in width (Bewsell et al., 2007; Howard-Williams et al., 2010; Gunn and Rutherford, 2013). By excluding (dairy) cattle from channels and riparian zones, the contribution of riverbank and bed erosion to degraded *CLAR* has likely been mitigated and reduced over time (Trimble and Mendel, 1995; Hughes and Quinn, 2014)."

Line 896: it is assumed that there is no doubt that fencing has improved the clarity of NZ rivers, but this is only a possible explanation. No evidence is given in the manuscript. In the presented study no specific quantitative analysis on fencing and sediment concentrations has been conducted.

We have toned down the language here and on line 672. Line 896 now reads: "While riparian fencing has plausibly improved the clarity of NZ rivers, …" Note that we have also beefed up the section on line 672 with references (below) and data.

Gunn, A., and Rutherford, C.: The Dairying and Clean Streams Accord: Snaphot of Progress 2011/2012, Ministry for Primary Industries, Wellington, 12, 2013.

Hughes, A. O., and Quinn, J. M.: Before and After Integrated Catchment Management in a Headwater Catchment: Changes in Water Quality, Environmental Management, 54, 1288-1305, 10.1007/s00267-014-0369-9, 2014.

Trimble, S. W., and Mendel, A. C.: The cow as a geomorphic agent—A critical review, Geomorphology, 13, 233–253, 1995.

---

## Author Comment (AC3) · 14 Oct 2016

**Authors' Reply to Editor**

**Major issues:**

Novelty of the results: It is difficult to identify what is novel with this manuscript as compared to other published studies on water quality trends in New Zealand rivers (and in general). In the Introduction for example, the reader cannot find clear statements about findings from the NZ monitoring activities and the open question that need to be addressed (e.g., L. 79 – 82: what are the reported trends, what are open questions?). In which sense does this manuscript add to previously published work?
We realize that the novelty and contribution of our work was not made clear in the original Introduction. Thus, we have revised the Introduction considerably to emphasize water quality response to change in land use *intensity*. We also changed the title of the paper to: **River water quality changes in New Zealand over 26 years (1989 – 2014): Response to land use intensity**. We have pasted the last four paragraphs of the new Introduction below, which lays out what has been done in terms of multi-catchment land-water relationships, our current lack of understanding of land use intensity, and our novel contribution:

[revised manuscript text omitted]

L. 40: This statement does hold in its generality – many important human activities (e.g., arts, science etc.) are not reflected in water quality.
Sentence revised to say: "River water quality reflects multiple activities and processes within its catchment, including geomorphic processes, vegetation characteristics, climate, and anthropogenic land uses."

L. 55 – 64: These sentences evoke the impression that mitigation has only focused on point-source pollution and that causal understanding about diffuse pollution is lacking. However, abating diffuse pollution has been pursued in many countries for several decades and many essentials are known about the drivers of diffuse pollution. Corresponding monitoring activities have been implemented and yield insight about success and short-comings of such programs (see (Kronvang *et al.* 2008) as just one arbitrary example). Your wording should better reflect the international state-of-the-art.
We added a sentence before the last sentence that reads: "Although considerable effort has been directed at monitoring and reducing diffuse pollution with some success, the legacy of pollutants from various land uses remains (Boesch, 2002; Kronvang et al., 2008)."
*Boesch, D. F.: Challenges and opportunities for science in reducing nutrient over-enrichment of coastal ecosystems, Estuaries, 25, 886-900, 10.1007/bf02804914, 2002.*

L. 65 – 67: I think this statement does not hold true (see comment above).
Based on the literature, we do believe that *most* studies on land use effects on water quality have been modeling studies, but we have toned down the language here and replaced 'most' with 'many'. We have also added the condition of "bracketing land use change" (see revised Introduction above).

L. 73 – 74; 87 – 88; 124 - 125: I doubt whether these statement hold true: There are many countries that have well-designed monitoring programs that run for several decades already. Some of them may also have more advanced sampling strategies than what you describe in this manuscript. In Switzerland for example, the national monitoring program starting in the 1970s for some parameters provides decade-long time-series of flow-proportional samples at weekly or bi-weekly resolution instead of monthly grab samples (Zobrist & Reichert 2006; Zobrist 2010). In the UK, some time-series date back in time over 100 yrs and make clear links between water

quality and land use (Howden *et al.* 2011). Your statements should better reflect what others have been doing for quite some time.

Thank you for bringing the Zobrist & Reichert (2006) reference to our attention. It fits well with this paragraph and we have added it. We have also added this reference to the last sentence of the previous paragraph because that was one of their concluding recommendations: "Comparisons of many diverse catchments is probably most useful to advance understanding of broad-scale land-water relationships (Zobrist and Reichert, 2006)." (We would note though, that although flow-proportional sampling such as that by these authors is valuable for load estimation, it is not appropriate for state-of-environment monitoring which needs to be random or pseudo-random (e.g. regular) with time.) The other references you suggest do not fit because this paragraph only covers studies that have compared water quality to land use for multiple catchments. To make this clear on line 73, we replaced the word 'riverine' with 'multi-catchment.' Between now and the final revision, we will keep looking for more of these types of studies. In the statements on lines 87-88 and 124-15, we say 'one of …', which is true. NZ's dataset of consistently monitored (same agency using the same protocols) monthly water quality for 15 variables for 77 catchments over 26 years is rivaled by only a few other datasets. We do not claim that NZ has *the* longest or *the* most comprehensive water quality dataset, and have modified the wording in the ms to emphasize the NRWQN's longevity, consistency and comprehensiveness, while not making any global claims for first ranking.

L. 161: P retention: please provide a reference for this method.

The Webb and Wilson (1995) reference at the beginning of the paragraph describes the methods for all soil properties used in our study, including P-retention. But we have added another reference (Saunders, 1965) that better describes the concept, process, and patterns across New Zealand.

*Saunders, W. M. H.: Phosphate retention by New Zealand soils and its relationship to free sesquioxides, organic matter, and other soil properties, N. Z. J. Agric. Res., 8, 30-57, DOI: 10.1080/00288233.1965.10420021, 1965.*

L. 223 – 228: Do you have ground truth data?

We have added the following sentence to the end of the paragraph: MODIS disturbance data were visually validated against 7500 random pixels from Landsat imagery and corresponding 15 high resolution Orbview-3 and Ikonos images. The overall accuracy of the disturbance index based on Landsat data was 98%.

Table 1: Give the temporal resolution of the data sets.

In the caption, and in text describing the NRWQN, we state that these are monthly data. We have added that the protocol was to take 'grab' samples to characterize a site; there was no compositing over time or space.

Table 8: For judging the quality of the regressions one should see scatter plots of the actual data.

We do show scatterplots for the primary and secondary explanatory variables in Figure 4. We now mention this in the caption to Table 8.

Fig. 4: The plots reveal pronounced heteroscedasticity. How did you deal with this issue?

We mention this on line 507. We did not correct for this in any way, but we did identify some of the outliers on Fig. 4 and explain why these occur in the Discussion section. For example, RO2 has high TP and DRP despite low $SUD_{cattle}$ because it has high plantation forestry coverage (line 836).

---

## Author Response (AR1)

**Authors' Reply to Editor's second round of comments**

**Reviewer # 2**:
**Reviewer**: "Furthermore the study primarily focusses on suspended sediment driven water quality constituents like suspended sediment concentration, total nitrogen and total phosphorus but the analysis is restricted to monthly data, hence the most important short term events with high concentrations of the abovementioned compounds are not considered in the study."
**You responded**: "Furthermore, although targeted sampling of high flow events is very relevant for load estimation of particle-related contaminants, it is not appropriate for state-of-environment monitoring like NRWQN (e.g. Davies-Colley et al. 2011; cited), covering both dissolved and particulate constituents, for which random or pseudo-random (e.g. regular monthly as in the NRWQN) sampling is most appropriate."
**My assessment**: This argument is only partially convincing. High flow events also matter and are part of the state of the environment. However, low flow conditions (generally well characterized by sufficiently long time series of grab samples and median values) and high flow events have different ecological relevance (see Stamm, Jarvie & Scott 2014 for illustration of my argument). Hence, what your sampling scheme and statistical methods reproduces are conditions that prevail for most of the time in the streams and what the organisms living there experience for most of the time. Therefore, it makes sense to look at these metrics and analyse these trends. However, this is not an argument to disqualify the critique that high flow events are not/only poorly captured by the sampling and statistical strategy. High flow events may be essential for water quality assessment (depending of parameters). This holds on the one hand if you think about downstream systems (including estuaries); on the other hand, it may also be essential for in-stream processes: if you wish to understand for example bed sediments in streams with all their ecological relevance you will hardly be able to do so by only knowing what happens during low flow conditions. Much research has been done illustrating how grab samples may severely underestimate loads of compounds entering streams predominantly during high flows and how difficult it may be to detect trends in time with such a sampling strategy (e.g., Moosmann *et al.* 2005). You have to explicitly mention these aspects in your manuscript and you have to make it clear to the reader what the results actually represent and what not. In the current version this is completely lacking: there is no discussion about sampling effects on results for example.
**Authors' response**: We added a paragraph at the beginning of the Discussion to acknowledge that we did not capture large floods, the resulting potential uncertainty, and justification for our ability to assess median conditions and trends. We included the references you mention, but did not include any discussion on in-stream processes because it was beyond the scope of our study.

**Reviewer**: "Furthermore the manuscript is very long (41 pages text only) and not very specific including repetitions."
**You responded**: "The manuscript is long because of our comprehensive coverage of both spatial and temporal effects of land use on a wide range of river water quality variables in complex large catchments. Arguably, the paper could be split into two manuscripts, but we feel it will have a greater impact as one paper. Further, an understanding of temporal effects is necessary in order to explain some of the spatial effects, and vice versa. We do not understand the comment 'not very specific.' We did a lot of investigation on land use practices and processes that were responsible for the patterns and relationships we observed. Maybe the reviewer is referring to our scale of analysis: catchment-scale. On line 95, we state: "Most of our analyses were performed at the catchment scale because it integrates the spatiotemporal changes that are reflected in our water quality measurements, it is the appropriate scale to analyze diffuse pollution, and it is the most appropriate spatial management unit (Howard-Williams et al., 2010)."

**My assessment**: I had a fresh look at the manuscript by reading it carefully once more and very much agree with the reviewer. Being comprehensive is nice but if this leads to a lengthy manuscript that distracts the reader from the essentials it has to be avoided. Being concise is beneficial to both – authors and readers: to the reader because he or she gets the relevant novel information as quickly and clearly as possible, to the author because the readers will like the paper more, which increases the probability of being cited later on. Given the fact that Reviewer # 2 called for a clear focus of the manuscript and your suggested focus and effects of land use *intensity*, I strongly recommend that the result and discussion really concentrate on this aspect. Below, I list some examples of lengthy and repetitive sections and paragraphs that should be avoided:

**Authors' response**: We address all of these separately below. We have also gone through the paper carefully and removed other parts of the ms we did not feel was essential. Overall, we have shortened the manuscript considerably.

- L. 281 – 283, 740 – 741 (and elsewhere): These details on which catchments shows what is hardly relevant for the general reader. Only indicate such details if they illustrate an aspect a reader cannot not understand otherwise and which is essential for the manuscript.
Lines 281-283 were removed and the preceding two sentences were combined and condensed. Lines 740-741 were also removed.

- L. 297 – 312: This can be shortened.
We shortened this section by about 3 lines.

- L. 417 – 429: These two paragraphs do not focus on changes in land use intensity, which should be the focus of the manuscript. Hence, they can be massively shortened or even skipped.
This paragraph reports changes (or lack thereof) in important water quality variables from 1989 to 2014, which is a part of the study. Further, when talking about changes in nutrients, it is good to have information on water temperature, DO, conductivity, and pH for context. Thus, we have kept this section.

- L. 450 – 457: This paragraph can also be shortened without loss of information essential to the manuscript.
We shortened this paragraph, removing 4 sentences.

- L. 520 – 547: These two paragraphs do not focus on changes in land use intensity, which should be the focus of the manuscript. Hence, they can be massively shortened.
These paragraphs characterize the states and trends of water quality in NZ rivers, which is a key part of this paper. This Discussion on differences between lowland and upland catchments also sets up the next section on "The role of physiography in dictating land use intensity across NZ". We feel it is important to leave this section in the paper.

- Section 5.2 (L. 597 – 634): This is not the focus of this manuscript because you focus on land use intensity. You can shorten this part substantially without loss of information.

We changed the title of this section to: The role of physiography in dictating land use intensity across NZ. This section is important because it makes the connections between physiography and land use intensity, which is then expanded on in the next section as regards river water quality.

- L. 536: This is repetitive and can be skipped (L. 405 – 406, L. 520, Tab. 5)
We deleted L536.

- L. 638 – 651: Repetitive, skip.
This entire section was removed.

- L. 667: This is repetitive and can be skipped.
Removed as suggested.

- L. 718 – 720: This is repetitive and can be skipped (L. 507, 654!).
L654 refers to high-producing grasslands, while L718-720 refers to plantation forests. To shorten this sentence, we removed the explanations of negative and positive relationships.

- Section 5.3.4 (L. 749 – 784): Lengthy descriptions of land use effects related to land use categories with little relevance for this study. Without loss of information you can either skip or shorten to 2 – 3 sentences at maximum.
We have removed this entire section.

- L. 846 – 855: This explanation of an outlier in the data set is superfluous. You already presented in quite some detail the reasons for a first outlier (L. 834 – 845). This first case may be included as a proof of concept for what the data set allows for, the second does not add any general insight. The general scientific audience is not interested in all details that may be of relevance for local or regional water managers.
We removed the 2nd paragraph, and added the mentioning of RO3 as another outlier to the previous paragraph.

- L. 898 – 902, 915 – 923: The content of the two respective paragraphs is very redundant and can be significantly shortened without loss of information.
We combined and condensed these two paragraphs.

- L. 907 – 911: This reads like a political statement for an NZ-internal audience and has no link to the actual content of the manuscript.
We removed these sentences.

**Reviewer**: "… and some conclusions are made without clear evidence."
**My observation**: Along the line mentioned in general terms I stumbled across two points I'd like to mention here:
i) On L. 87 – 88 you make a bolt statement about the NZ water quality monitoring program ("it has one of the longest comprehensive national water quality datasets in the world"). I suggest that you back this with some information that supports this statement.

We changed the wording to say: it has a long, consistent, and comprehensive national water quality dataset.

ii) ii) Your conclusion regarding the possible effects of increasing water clarity (L. 581 – 595). This is an interesting point but you do not provide evidence for your statement that "when combined with increasing nutrients, warmer water, and lower flows, the perfect recipe for toxic algae blooms is created." (L. 581 – 583). You cite (McAllister, Wood & Hawes 2016) but these authors seem to contradict your statement by claiming: "While quantitative data on sedimentation rates in rivers is lacking at a national scale, increasing land use intensification and forestry are likely to result in increased sediment in rivers, which may be partly responsible for observed rise in Phormidium proliferations." (McAllister, Wood & Hawes 2016, p. 292). Please provide references that support your statement.

McAllister's point in this paragraph is that because most NZ freshwaters are strongly P-limited, increased dissolved reactive phosphorus (DRP; which has a strong association with sediment) leads to Phormidium proliferations (see 3$^{rd}$ sentence in their Conclusions for clarification). McAllister acknowledges light-limitation on p.287. We have added references to support our claim on L581: Dodds and Welch, 2000; Hilton et al., 2006. We also used these references to back up our claim in the final sentence of this paragraph.

**Editor comments**:

Already in my initial comments I asked for scatter plots displaying the relationships between discharge and water quality parameters. You argued that this would be an overkill. As a consequence, the entire flow normalization (L. 137) that you do as the first step in the data processing is basically hidden from the reader. As a consequence, your entire section 4.3.1 reports data without showing actual data although you qualitatively describe concentration-discharge relationships. This is not satisfactory and could be easily alleviated. If you combine related water quality parameters (e.g., total P, DRP etc.) together in one plot, you can display these concentration-discharge relationships with 4 to 5 matrices of 9x9 plots for all 77 sites. You have already now plot matrices of 16 x 16 in the Supplementary material.

Therefore, providing scatterplots of the concentration-discharge relationships with the LOESS functions is doable and will provide the reader with essential access to real data. This will improve the quality of the manuscript substantially and will make the entire process of data processing much more transparent. I strongly suggest that you add this information.

Now that the paper is focused on land use intensity, we have removed section 4.3.1 (L368-403) entirely. Flow-normalization using LOESS is a commonly used technique and we don't think the 770 plots (with 312 data points each) would add any value to the paper, especially since section 4.3.1 has been removed. If we combined two variables on one plot, that would still be 385 plots, but now with 624 data points each, which would look like a buckshot. For the Editor's benefit, we have added a supplement below that shows the LOESS plots for just one catchment, AK1. We would have to produce this group of 10 plots 76 more times to capture all the catchments.

Because your focus is on the effects of intensity change you might consider to motivate the issue by actually including a figure in the main text that illustrates that land use did hardly change between 1990 and 2012 but that intensity did (based on Tab. 2 & 3 in the Supplementary Material).

This is a great idea. We have added this figure to the ms, now Figure 2.

**Author supplement 1: LOESS plots for catchment AK1**

[revised manuscript text omitted]

---

## Editor Decision (ED1)

Dear Dr. Julian

HESSD-Manuscript "River water quality changes in New Zealand over 26 years (1989–2014): Response to land use and land disturbance" (HESS 2016-323).

Thank you for providing the answers to the comments by the reviewers . I carefully went through the list of your replies. You responded to all points that were mentioned. However, not all of your answers are convincing and I list the critical aspects below:

**Reviewer # 2**:

Reviewer: "Furthermore the study primarily focusses on suspended sediment driven water quality constituents like suspended sediment concentration, total nitrogen and total phosphorus but the analysis is restricted to monthly data, hence the most important short term events with high concentrations of the abovementioned compounds are not considered in the study."

You responded: "Furthermore, although targeted sampling of high flow events is very relevant for load estimation of particle-related contaminants, it is not appropriate for state-of-environment monitoring like NRWQN (e.g. Davies-Colley et al. 2011; cited), covering both dissolved and particulate constituents, for which random or pseudo-random (e.g. regular monthly as in the NRWQN) sampling is most appropriate."

My assessment:  This argument is only partially convincing. High flow events also matter and are part of the state of the environment. However, low flow conditions (generally well characterized by sufficiently long time series of grab samples and median values) and high flow events have different ecological relevance (see Stamm, Jarvie & Scott 2014 for illustration of my argument). Hence, what your sampling scheme and statistical methods reproduces are conditions that prevail for most of the time in the streams and what the organisms living there experience for most of the time. Therefore, it makes sense to look at these metrics and analyse these trends. However, this is not an argument to disqualify the critique that high flow events are not/only poorly captured by the sampling and statistical strategy. High flow events may be essential for water quality assessment (depending of parameters). This holds on the one hand if you think about downstream systems (including estuaries); on the other hand, it may also be essential for in-stream processes: if you wish to understand for example bed sediments in streams with all their ecological relevance you will hardly be able to do so by only knowing what happens during low flow conditions.

Much research has been done illustrating how grab samples may severely underestimate loads of compounds entering streams predominantly during high flows and how difficult it may be to detect trends in time with such a sampling strategy (e.g., Moosmann *et al.* 2005).

You have to explicitly mention these aspects in your manuscript and you have to make it clear to the reader what the results actually represent and what not. In the current version this is completely lacking: there is no discussion about sampling effects on results for example.

Reviewer: "Furthermore the manuscript is very long (41 pages text only) and not very specific including repetitions."

You responded: "The manuscript is long because of our comprehensive coverage of both spatial and temporal effects of land use on a wide range of river water quality variables in complex large catchments. Arguably, the paper could be split into two manuscripts, but we feel it will have a greater impact as one paper. Further, an understanding of temporal effects is necessary in order to explain some of the spatial effects, and vice versa. We do not understand the comment 'not very specific.' We did a lot of investigation on land use practices and processes that were responsible for the patterns and relationships we observed. Maybe the reviewer is referring to our scale of analysis: catchment-scale. On line 95, we state: "Most of our analyses were performed at the catchment scale because it integrates the spatiotemporal changes that are reflected in our water quality measurements, it is the appropriate scale to analyze diffuse pollution, and it is the most appropriate spatial management unit (Howard-Williams et al., 2010)."

My assessment:  I had a fresh look at the manuscript by reading it carefully once more and very much agree with the reviewer. Being comprehensive is nice but if this leads to a lengthy manuscript that distracts the reader from the essentials it has to be avoided. Being concise is beneficial to both – authors and readers: to the reader because he or she gets the relevant novel information as quickly and clearly as possible, to the author because the readers will like the paper more, which increases the probability of being cited later on.

Given the fact that Reviewer # 2 called for a clear focus of the manuscript and your suggested focus and effects of land use *intensity*, I strongly recommend that the result and discussion really concentrate on this aspect.  Below, I list a some examples of lengthy and repetitive sections and paragraphs that should be avoided:

- L. 281 – 283, 740 – 741 (and elsewhere): These details on which catchments shows what is hardly relevant for the general reader. Only indicate such details if they illustrate an aspect a reader cannot not understand otherwise and which is essential for the manuscript.
- L. 297 – 312: This can be shortened.
- L. 417 – 429: These two paragraphs do not focus on changes in land use intensity, which should be the focus of the manuscript. Hence, they can be massively shortened or even skipped.
- L. 450 – 457: This paragraph can also be shortened without loss of information essential to the manuscript.
- L. 520 – 547: These two paragraphs do not focus on changes in land use intensity, which should be the focus of the manuscript. Hence, they can be massively shortened.
- Section 5.2 (L. 597 – 634): This is not the focus of this manuscript because you focus on land use intensity.  You can shorten this part substantially without loss of information.
- L. 536: This is repetitive and can be skipped (L. 405 – 406, L. 520, Tab. 5)
- L. 638 – 651: Repetitive, skip.
- L. 667: This is repetitive and can be skipped.
- L. 718 – 720: This is repetitive and can be skipped (L. 507, 654!).
- Section 5.3.4 (L. 749 – 784): Lengthy descriptions of land use effects related to land use categories with little relevance for this study. Without loss of information you can either skip or shorten to 2 – 3 sentences at maximum.
- L. 846 – 855: This explanation of an outlier in the data set is superfluous. You already presented in quite some detail the reasons for a first outlier (L. 834 – 845). This first case may be included as a proof of concept for what the data set allows for, the second does not add any general insight. The general scientific audience is not interested in all details that may be of relevance for local or regional water managers.

- L. 898 – 902, 915 – 923: The content of the two respective paragraphs is very redundant and can be significantly shortened without loss of information.
- L. 907 – 911: This reads like a political statement for an NZ-internal audience and has no link to the actual content of the manuscript.

Reviewer: "… and some conclusions are made without clear evidence."

My observation: Along the line mentioned in general terms I stumbled across two points I'd like to mention here:

i) On L. 87 – 88 you make a bolt statement about the NZ water quality monitoring program ("it has one of the longest comprehensive national water quality datasets in the world"). I suggest that you back this with some information that supports this statement.

ii) ii) Your conclusion regarding the possible effects of increasing water clarity (L. 581 – 595). This is an interesting point but you do not provide evidence for your statement that "when combined with increasing nutrients, warmer water, and lower flows, the perfect recipe for toxic algae blooms is created." (L. 581 – 583). You cite (McAllister, Wood & Hawes 2016) but these authors seem to contradict your statement by claiming: "While quantitative data on sedimentation rates in rivers is lacking at a national scale, increasing land use intensification and forestry are likely to result in increased sediment in rivers, which may be partly responsible for observed rise in Phormidium proliferations." (McAllister, Wood & Hawes 2016, p. 292). Please provide references that support your statement.

**Editor comments**:

Already in my initial comments I asked for scatter plots displaying the relationships between discharge and water quality parameters. You argued that this would be an overkill. As a consequence, the entire flow normalization (L. 137) that you do as the first step in the data processing is basically hidden from the reader. As a consequence, your entire section 4.3.1 reports data without showing actual data although you qualitatively describe concentration-discharge relationships. This is not satisfactory and could be easily alleviated. If you combine related water quality parameters (e.g., total P, DRP etc.) together in one plot, you can display these concentration-discharge relationships with 4 to 5 matrices of 9x9 plots for all 77 sites. You have already now plot matrices of 16 x 16 in the Supplementary material.

Therefore, providing scatterplots of the concentration-discharge relationships with the LOESS functions is doable and will provide the reader with essential access to real data. This will improve the quality of the manuscript substantially and will make the entire process of data processing much more transparent. I strongly suggest that you add this information.

Because your focus is on the effects of intensity change you might consider to motivate the issue by actually including a figure in the main text that illustrates that land use did hardly change between 1990 and 2012 but that intensity did (based on Tab. 2 & 3 in the Supplementary Material).

**Overall recommendation:**

I recommend that you revise the manuscript based on your response to the reviews and that you pay due attention to the critical points that the referees and myself have raised.

Sincerely

Dr. Christian Stamm
Editor HESS
Environmental Chemistry
Eawag
Ueberlandstr. 133
CH-8600 Dübendorf, Switzerland

References:

McAllister, T.G., Wood, S.A. & Hawes, I. (2016) The rise of toxic benthic Phormidium proliferations: A review of their taxonomy, distribution, toxin content and factors regulating prevalence and increased severity. *Harmful Algae,* **55,** 282–294.

Moosmann, L., Müller, B., Gächter, R., Butscher, E., Herzog, P. & Wüest, A. (2005) Trend-oriented sampling strategy and estimation of soluble reactive phosphorus loads in streams. *Water Resources Research,* **41,** W01020, doi:01010.01029/02004WR003539.

Stamm, C., Jarvie, H.P. & Scott, T. (2014) What's more important for managing phosphorus: Loads, concentrations or both? *Environmental Science & Technology,* **48,** 23–24.

---

## Author Response (AR2)

**Authors' response to Editor's and Reviewers' comments**

**Editor comments**:

I also acknowledge that the manuscript was seriously revised and has substantially be improved. Nevertheless, I still see two major deficiencies. The first is the length of the manuscript, which is partially due to the fact that the results and discussion contain side stories that are not essential to the paper and distract the reader (see detailed comments below). Please note that what I call side stories here are not necessarily unimportant per se, but they are superfluous in the context of this paper.

We have shortened the paper according to your comments below and in many other places. We removed most of the catchment identifiers throughout the ms, except in parts of the Discussion where it is necessary and those that relate directly to Figure 5 where we point out specific catchments. We removed a couple of the 'side stories,' but kept the ones we felt were essential to the story of how river water quality responds to land use intensity, which requires a physiographic context. We justify these instances below.

The second deficiency relates to the actual content and focus of the paper. In this revised version you put the emphasis on land use intensity – at least in the title and the abstract. However, the actual result section consists only of one out of six subsection that specifically addresses land use intensity (sub-section 4.3) and sub-section 4.6 on multivariate is only about correlation between land use and water quality. Land use intensity is not dealt with.

We respectfully disagree with this assessment that land use intensity is not covered throughout the ms. We wrote the Introduction in a manner that first summarizes what has been done in land use-water quality relationships (i.e. observations) in order to develop and justify our hypothesis that land use intensity is a better predictor of water quality. Even when we discuss land uses, we put them within the context of land use intensity. For example, section 4.2 distinguishes between land uses that are intensively managed (high-producing grasslands and plantation forests) versus those that are not intensively managed (low-producing shrub/grassland and non-plantation forest). Further in section 4.2, we have to point out that areal coverage has not changed much so that we can make the case for land use intensity driving the changes in water quality. Another example is section 4.5 where we relate water quality to intensively managed grasslands, stock unit densities, and disturbance. We acknowledge that some small sections do not deal specifically with land use intensity, but these are mostly cases where we need to characterize physiographic variability in order to understand water quality changes with land use intensity in different environments. For example, water quality in catchments with high soil phosphate retention responds differently to agricultural land uses. Further, we make the connections between physiography and land use intensity (section 4.3). Section 4.6 does address land use intensity, indirectly when it reports relationships with high-producing grasslands and directly when it reports that stock unit density was the primary predictor for all four nutrient variables. Overall, keep in mind that we are covering both land use intensity and its effect on water quality. This is why the first section of the Discussion (5.1) covers water quality relationships and patterns. We relate land use intensity to physiography in section 5.2 and then bring everything together in section 5.3 which is all about land use intensity (i.e. livestock densities, fertilizer use, disturbance. This section is by far the longest section of the ms, the concluding section, and the most important. Our final point in this regard is that this is a true 'earth systems science' study (the focus of this journal) where we not only look at relationships between land use intensity and water quality, but take into account all aspects of the earth system such as climate, topography, soils, vegetation, and human-environment interactions. Nevertheless, we have revised some parts of the ms to make our discussion of land use intensity more explicit. Examples include 2$^{nd}$ sentence of section 4.2, 2$^{nd}$ paragraph of section 5.1, L27 of Abstract, and 1$^{st}$ sentence of section 5.3.3.

In my previous comments I asked explicitly for a clear focus on intensity: "Given the fact that Reviewer # 2 called for a clear focus of the manuscript and your suggested focus and effects of land use intensity, I strongly recommend that the result and discussion really concentrate on this aspect." When evaluating the actual content of the results section of the revised version one has to state that the largest part does not deal with land use intensity. This discrepancy between title/abstract on the one hand and the actual content on the other hand, is annoying and evokes the impression that the revisions were only superficial. I can only accept this manuscript for publication if this aspects is seriously improved. I am convinced that a stronger focus will automatically lead to a shorter manuscript improving on the length issue.
See our reply above, and when reviewing our ms, please keep in mind that land use intensity includes livestock densities, fertilizer use, and disturbance, which are the focus of the ms. This is made clear in the 2$^{nd}$ sentence of the abstract and the Introduction (L90).

Detailed comments (Line numbers refer to the version with track changes of the Author's Response):
My line numbers refer to the version submitted on 11-04-2016 without track changes.

L. 15: Interdependencies and feedbacks between what? Please clarify.
between land use and water quality. We rephrased.

L. 27: This statement seems to be contradicted by L. 31 – 34. Rewording might help to avoid the confusion for a reader.
We clarified on L27 that it was median visual water clarity. L31 refers to temporal changes.

L. 31 – 34: Where do I find this information in the result section?
Table 6 and L422.

L. 70: cause-effect relationships between what?
We added "… between land use and water quality".

L. 75 – 77: One can easily argue against this statement because in small catchments the actual transfer may be dependent on very site-specific conditions affecting connectivity, which are difficult to incorporate into existing model approaches. I suggest to tone down the statement or to provide references that actually support your statement.
We removed the word 'small' and added 'heterogeneous' to large catchments.

L. 264: How was this visual validation carried out? Please show results of the validation in the SI.
The methods and results for validation are detailed in de Beurs et al. (2016), which we state earlier in this paragraph. The results of this validation are in Table 4 of that article, so no need to reproduce here. We feel the term 'visual validation' of disturbance against high-resolution imagery is self-explanatory. We looked at the image to see if the ground was disturbed (i.e. no vegetation).

L. 282: These trends were flow corrected?
The trends were calculated on flow-normalized water quality data. We added this clarification.

L. 300 – 301: Skip this reference – that a 1% change per year results in a > 10% change in a decade is simple maths.
Reference deleted.

L. 313 – 315: These details can be skipped for the sake of brevity (unless these catchments have a particular relevance which is not reported in the manuscript). For the general readers these abbreviations are just meaningless. This is of course different for people managing the specific areas. However, this paper is written for a general audience and not for local authorities.
We removed the catchment identifiers.

L. 328 – 341: This very descriptive part has to be shortened considerably.
We removed the catchment identifiers and other nonessential information, but most of the paragraph needs to stay because it characterizes the coverage of land use of our study area, which the reader needs to know in order to understand the extent of land use intensity. This is why we distinguish between intensively-managed grasslands and grasslands not intensively managed. Likewise, we distinguish between non-plantation forests and the intensively-managed plantation forests. We revised the paragraph to make this clear. Further, it is important to characterize the minimal coverages of the other land uses at the end so that the reader understands that urban, cropland, water/wetlands, and barren/other land uses have a relatively minor influence on water quality compared to the agricultural uses.

L. 576 –604: What is the novelty here? This very descriptive part has to be shortened considerably.
We revised the beginning of this section to relate explicitly to land use intensity. We then condensed these two paragraphs down to one, much shorter paragraph.

L. 854: Where can the results be seen?
These results are reported in the 3rd paragraph of section 4.3.

L. 861: You provide a reference here – so what is actually novel here. Skip was simply represents a repetition of previous findings.
We removed this detailed example, only keeping the reference that documents this process.

L. 881 – 882: Where are the data? Why is this not yet presented in the result section? Or did I miss it?
We added a reference to Supplement Tables 3 & 4, but this result is presented at the end of section 4.3 (L390).

L. 891 – 893: This argument is not convincing. What shall be the actual mechanism that prevents a high concentration to increase further?

We looked back over these data and found that the reason was "because of the extreme monthly variability in river nitrogen concentrations, possibly due to livestock rotations, fertilizer applications, and precipitation events." We revised this sentence accordingly. I have also pasted the plot below.

[Figure]

L. 897 – 907: This part distracts from the main story. Skip.
L. 909 – 918: Again, this explanation of outliers distracts from the main story. Skip.
This paragraph is part of our main story, that areal coverage alone does not explain water quality. Hence our finding: "Our high temporal-resolution analyses of disturbance showed that even though this catchment is mostly indigenous forest, intense disturbances on small proportions of developed land can have a considerable impact on water quality." Disturbance is one of our metrics of land use intensity, and this paragraph provides key examples why this disturbance metric is useful and why land use intensity is a better predictor of water quality in some cases. Also note that we did shorten this paragraph in the previous revision.

L. 919 – 934: Shorten.
We shortened this paragraph some.

**Reviewer #1**
Specific suggestions
L17-18 SRA '...land use intensity, defined here as the inputs of fertilizer and livestock density, and land disturbance via vegetation removal, is a better predictor of water quality impact.'
We kept our original wording because this suggested wording makes it sound like land disturbance is separate from land use intensity.

L23-26 SRA ' We analyzed spatial and temporal changes in livestock density and bare soil at the catchment-scale, as well as fertilizer inputs at the national scale.'

We feel it is necessary here to specify what causes the land disturbance (i.e. grazing and forest harvesting). Just saying bare soil could imply desert, barren land, or cultivated crops.

L28 SRA '...for four nutrient variables (a,b,c,d), however, ...'
Nutrient variables added as suggested.

L31-34 I expected to read here an explanation of why high producing pastures were positively related to water clarity. The reasoning is here but is presented as a separate finding rather than an explanation of the finding presented in L27. Please consider rephrasing.
The statement in L27 is a spatial comparison; whereas, L31 is a temporal comparison. These are two separate analyses. Hopefully our addition of 'median visual water clarity' clears this up.

L38 Please quantify 'some' as a percentage or proportion.
This would require several sentences given that we would have to address five different variables, and rivers that improved in TN did not necessarily improve in TP, or clarity, and so on. Because of the reviewers' insistence that we shorten the ms, we did not expand this sentence to reflect proportions of rivers that improved for all five variables.

L36 So is increased cattle density the cause of both improved water clarity and increased NOX?
Now we see where part of the confusion arose in the comment before last. To clear up this confusion, we added that the relationship is inverse: "median visual water clarity was best predicted inversely by …"

L38 Please quantify 'some' as a percentage or proportion. Re 'nutrients' is this P and N or just one of these nutrients? Please be specific.
As mentioned above, this would require several sentences given that we would have to address five different variables, and rivers that improved in TN did not necessarily improve in TP, or clarity, and so on. Because of the reviewers' insistence that we shorten the ms, we did not elaborate here.

L63 SRA '...pollution from fine sediments, nutrients...'
Revised as suggested. Note that this is one of the sentences we shortened.

L92-93 Consider the wording I suggest for the Abstract
We kept our original wording because this suggested wording makes it sound like land disturbance is separate from land use intensity.

L94 Suggest delete 'areal coverage' and replace with 'land use alone', because you look at areal coverage of land use intensity over time.
Revised as suggested.

L96-97 Reason 1 holds for land use alone also, so I suggest the added complexity of studying land use intensity effects is just reason 2.
These reasons were described in the Kuemmerle and Erb references provided. For one example that justifies reason 1, think about the land use of pasture. The relationship between pasture and landscape variables such as soil type is relatively straightforward; however, the interactions among different livestock densities (i.e. land use intensity), pasture, and soil type is more complex and depends on other interacting variables. The key word in reason 1 is 'multidimensional;' think of land use intensity as adding another dimension, and thus another level of complexity/uncertainty.

L100 Please add a reason for 'decades pasture area has decreased' (eg due to conversion to ?) and specify which livestock have increased in density.
Both of these details are results from our analyses and thus presented in the Results section. We are only introducing the concept here.

L106-07 I expect that in some cases, such as conversion from sheep to dairy cattle pasture production, that land use change and land use intensity change are correlated. So some of these other analyses probably have addressed land-use intensity change if not directly, by proxy. Please comment.
These previous studies only looked at areal coverage of pasture (or high-producing grasslands); they did not look at changes in cattle vs sheep like we did. This is one of the reasons our study is novel.

L110 'one of the highest rates' needs a reference, or rephrasing to 'a high rate'
We added a reference, the same one we use in the Conclusions to make the same point.

L114 SRA 'variables that extend over'
Revised as suggested.

L118 re 'specifically livestock density and land disturbance'- and not fertiliser inputs as described earlier?
This statements refers to analyses of the 77 catchments. Fertilizer inputs were only assesses at the national level.

L125 catchment scales are appropriate scales for some levels of management, but not others. Field- or stream reach- scale management is appropriate for managing catchment scale outcomes in many cases. Please consider rephrasing.
We shortened this sentence and revised accordingly.

L206 Re 'intensity, and disturbance data' Earlier disturbance was defined as a sub-category of land use intensity, so please rephrase this subheading.
Revised.

L229-230 I don't follow why bare soil would be a proxy for high-producing grasslands. see below
L235 How does stock unit density relate to livestock impact on land disturbance and/or bare soil? THis seems out of place because this section describes how livestock density was calculated and the following paragraph describes how land disturbance was calculated for various land uses. Correctly in my view, SUD doesn't appear to be used to calculate land disturbance. Please consider deleting '..on land disturbance' or describe how SUD was linked to land disturbance, including the assumptions and basis for assumptions that were used to correlate stock density with bare soil.
We deleted '…on land disturbance' as suggested.

RE L240 How does the dataset of Ausseil for 2011 differ from that of StatsNZ and why was the Ausseil data needed? Please clarify
The Ausseil data has 1-ha resolution, which allows us to do spatial comparisons. The StatsNZ data is district-level data (which does allow precise spatial comparisons), but because it covers multiple years, it allows us to do temporal comparisons. To avoid confusion (on L245), we changed 'stock units per hectare' to 'stock unit densities.'

L243 I suggest 'intensity' rather than 'impacts' here.
Changed as suggested.

L328 To what time period do the subsequently reported land use extents refer - an average from 1990 to 2012?
In the methods (section 3.3), we have clarified that it is for the midpoint year of 2001. We also added in the 2$^{nd}$ sentence here that these coverages refer to the year 2001.

L365 SRA 'little over the period 1990-2012'
Revised as suggested.

L674-675 Can you provide an exemplar reference for this finding? To my knowledge, export of DRP to rivers in high Pret soils is usually driven by the tendency for runoff which is likely to be low in these free-draining allophanic soils. Other mechanisms for DRP transport to rivers in high Pret soils may be via preferential flow pathways (natural or artificial) or due to mobilisation of less readily fixed organic P forms and subsequent mineralisation into DRP at or near the point of delivery to the river.
It is common knowledge that "soils with high $P_{ret}$ require more P-fertilizer." And it is intuitive that the more P-fertilizer added to soils, the higher the DRP export to rivers through various pathways, as you mention here. And yes, there are both natural and artificial flow pathways in these intensively managed pastures, particularly the drainage canals. Thus, we have not added a reference for this intuitive logic.

**Reviewer #2**
Former Line 667: the correlation of 0.45 is still not high what also means it is not relatively high. Please correct this
This statement is no longer in the revised ms.

Please consider the editors comment on former lines 417-429 on shortening the manuscript as suggested
We shortened this paragraph.

Please shorten also the line 520-547 as suggested by the editor.
These suggestions on shortening the manuscript are mandatory!
We condensed these two paragraphs down to one, much shorter paragraph.

[revised manuscript text omitted]